# Improving tolerance to fluctuating light through adaptive laboratory evolution in the cyanobacterium *Synechocystis*

Theo Figueroa-Gonzalez[1,5], Weiyang Chen [1,5], Eslam M. Abdel-Salam [1], Daniel Štipl[2,3], Josef Komenda [2], Milena Zhivkovikj[1], Marcel Dann [1,4] & Dario Leister [1] ✉

Fluctuating light (FL) poses a challenge to cyanobacteria by disrupting photosynthesis and damaging photosystems. Although key FL tolerance components are known, their genetic enhancement remains unexplored. We evolve *Synechocystis* PCC 6803 under two FL regimes (one lethal to the starter strain, LT) in order to identify previously unknown adaptive alleles. Our analysis reveals 44 mutations, 28 of which impact proteins/RNAs. Mutations in Pam68 (PSII assembly) and Sll0518, present in all strains, enhance non-lethal FL tolerance in LT. Mutated Pam68 increased PSII abundance and activity. A gain-of-function mutation in RpaB (regulator of phycobilisome association B) significantly increases tolerance to both lethal FL and high-light conditions. This is associated with an increased PSI/PSII ratio and downregulation of light harvesting. In summary, our results suggest that adaptive laboratory evolution can simultaneously identify FL tolerance factors and their advantageous alleles. The identified point mutations rewire multiple protective responses by as yet unknown molecular mechanisms.

Cyanobacteria uniquely perform oxygenic photosynthesis using photosystems I and II (PSI, PSII), capturing light to drive water-splitting and create a proton gradient for ATP synthesis and $CO_2$ fixation[1,2]. While light is essential, high (HL) or fluctuating light (FL) induces photoinhibition[3,4]. Cyanobacteria often encounter HL and FL[5-7], yet FL tolerance is less understood than HL tolerance, which involves diverse adaptive mechanisms[6,8-21]. Studies on *Synechococcus elongatus* PCC 7942 and *Synechocystis* sp. PCC 6803 ("*Synechocystis*") reveal the importance of inorganic carbon and alternative electron pathways for FL tolerance[22-24]. Flavodiiron proteins[23,25,26], nitrogen assimilation[27,28], and thylakoid respiratory activity[29] enhance FL tolerance, and Fluctuating-light acclimation protein 1 (FLAP1)[30] also plays a role. Although more FL tolerance components likely exist, no genetic enhancement of cyanobacterial FL tolerance has been reported,

hindering the development of suitable production strains for FL-prone photobioreactors[31]. Similarly, improving flowering plant FL tolerance through genetic engineering has seen limited success, with a few exceptions in tobacco and soybean[32]. Further increases in acclimation potential may require an evolutionary approach entailing the identification of new FL tolerance factors and the evolution of advantageous alleles[33].

Previous adaptive laboratory evolution (ALE) studies on *Synechocystis* for increased HL tolerance have demonstrated the general accessibility of photosynthetic robustness to evolutionary improvement[34-36]. Therefore, we applied ALE to evolve alleles conferring FL tolerance in *Synechocystis*, resulting in the identification of distinct candidate mechanisms for tolerance to different types of light fluctuations. Mutations in Sll0518 and Pam68 conferred tolerance to

[1]Plant Molecular Biology, Faculty of Biology, Ludwig-Maximilians-Universität München, Munich, Germany. [2]Institute of Microbiology of the Czech Academy of Sciences, Centre Algatech, Třeboň, Czech Republic. [3]Faculty of Science, University of South Bohemia, České Budějovice, Czech Republic. [4]Bio-inspired Energy Conversion, Department of Biology, Technical University of Darmstadt, Darmstadt, Germany. [5]These authors contributed equally: Theo Figueroa-Gonzalez, Weiyang Chen. ✉e-mail: leister@lmu.de

moderate FL, while a RpaB mutation increased tolerance to both lethal FL and HL.

## Results

### Generation of FL-tolerant batch cultures and isolation and characterization of monoclonal strains

We adapted *Synechocystis* to tolerate FL using ALE, adapting previous methods for generating HL tolerance[35] and relying on the natural mutation rate of *Synechocystis*[33]. Two experimental protocols were designed to progressively increase FL intensity, modifying an existing FL regimen for *Arabidopsis thaliana*[37], which alternated between high (HL) and low light (LL) phases. The "FL0" protocol maintained the original 1-min HL / 5-min LL rhythm but increased light intensity: starting with 700 μmol photons $m^{-2}$ $s^{-1}$ for HL ($HL_{700}$) and 50 μmol photons $m^{-2}$ $s^{-1}$ for LL ($LL_{50}$), the intensities were gradually altered to $HL_{1200}$ and $LL_{12}$ ($FL0_{final}$), respectively (Table 1, Fig. 1a). While continuous $HL_{1200}$ is lethal to the non-adapted starter strain (LT)[35], the $FL0_{final}$ conditions allowed for recovery and growth during the LL phase, as demonstrated by the productive growth of the non-evolved LT strain under both FL0 and $FL0_{final}$ conditions (Fig. 1b).

The "FL+" protocol began with the same initial conditions as FL0 but progressively shortened the LL phase to 1 min. The final cycles alternated between 1 min at $LL_{12}$ and 1 min at $HL_{1200}$. These conditions ($FL+_{final}$) proved lethal for the LT strain (Fig. 1b). Both FL0 and FL+ protocols involved 20 selective cultivation cycles on triplicate batch cultures over 20 months.

After ALE, cultures showed phenotypic variations within and between triplicates (Fig. 1c–f, Source Data). Under their respective final FL conditions, FL+ strains exhibited higher growth rates and increased cell density compared to FL0 cultures, although with lower chlorophyll content. This suggests that the $FL+_{final}$ adapted strains not only tolerated the lethal light regime, but also made productive use of the greater light energy available in the FL+ condition compared to the FL0 strains, which received fewer total photons. The time-integrated $FL+_{final}$ photon flux (286%) corresponded to approximately three times that of $FL0_{final}$ (100%).

**Table 1 | The FL0 and FL+ selective regimes for FL-ALE**

| Condition | LL | HL | A | $t_{LL}$ | $t_{HL}$ | Cycles |
|---|---|---|---|---|---|---|
| FL0 | | | | | | |
| Initial | 50 | 700 | 650 | 5 | 1 | 5 |
| Intermediate | 50 | 1000 | 950 | 5 | 1 | 1 |
| | 50 | 1200 | 1150 | 5 | 1 | 1 |
| | 20 | 1200 | 1180 | 5 | 1 | 1 |
| Final | 12 | 1200 | 1188 | 5 | 1 | 12 |
| FL+ | | | | | | |
| Initial | 50 | 700 | 650 | 5 | 1 | 1 |
| Intermediate | 50 | 700 | 650 | 4 | 1 | 1 |
| | 50 | 700 | 650 | 3 | 1 | 1 |
| | 50 | 700 | 650 | 2 | 1 | 1 |
| | 50 | 700 | 650 | 1 | 1 | 1 |
| | 50 | 1000 | 950 | 1 | 1 | 1 |
| | 50 | 1200 | 1150 | 1 | 1 | 1 |
| | 20 | 1200 | 1180 | 1 | 1 | 1 |
| Final | 12 | 1200 | 1188 | 1 | 1 | 12 |

The light regimes implemented during the 20 propagation rounds of the two FL-ALE protocols (FL0 and FL+) are detailed. The parameters include: LL/HL, low/high light intensities, quantified in μmol photons $m^{-2}$ $s^{-1}$; A, Amplitude, calculated as HL - LL; $t_{HL/LL}$, duration of exposure to LL/HL expressed in min; and Cycles, number of propagation rounds. Throughout all experimental protocols, environmental conditions were standardized with an aeration rate of 100–150 mL of air per minute and a constant temperature of 23 °C.

The batch cultures were diluted, plated on solid media, and incubated under constant $LL_{20}$ (Fig. 1g), before individual clones were isolated, imaged and analysed with respect to their 'apparent' quantum yield of PSII (Fig. 1h). Note that analysing the quantum yield (Fv/Fm) in cyanobacteria can be problematic due to phycobilisome contribution to basal fluorescence and interference from respiration, and therefore this parameter is designated as apparent quantum yield or $Fv^-/Fm^-$[38,39]. The FL+ clones exhibited greater heterogeneity in $Fv^-/Fm^-$ values compared to FL0 clones (Fig. 1h, i). Four clones from each batch were selected to represent the quantiles of $Fv^-/Fm^-$, sampling a wide range of phenotypic diversity. FL0 isolates showed a homogeneous phenotype with dark green colour and $Fv^-/Fm^-$ of 0.39 ± 0.02. In contrast, FL+ clones displayed clear heterogeneity, with colours ranging from cyan to ochre and $Fv^-/Fm^-$ values between 0.19 and 0.53 (0.40 ± 0.10) (Fig. 1h, i; Source Data). Most FL0 isolates grew denser than LT under $FL0_{final}$ condition, while all FL+ clones survived and accumulated high cell densities under $FL+_{final}$ condition, which was lethal to the non-adapted LT (Fig. 1j; Source Data).

### Mutations in FL-tolerant monoclonal strains

Whole-genome analysis of the 12 FL0 and 12 FL+ monoclonal strains, using the LT strain from which these adapted strains were derived and the original motile *Synechocystis* PCC 6803 strain (designated "WT") as controls, yielded a mutation matrix revealing 412 mutations (234 in FL0 and 269 in FL+) absent in both LT and WT strains (Supplementary Fig. 1a, Source Data). The majority of these mutations (349 total, 201 in FL0 and 223 in FL+) were located within coding regions. Almost all (342 total, 198 in FL0 and 218 in FL+) were single nucleotide polymorphisms (SNPs), while seven (3 in FL0 and 5 in FL+) were insertions or deletions (InDels). Among the coding region SNPs, 277 (157 in FL0 and 182 in FL+) resulted in non-synonymous exchanges, modifying the amino acid sequence of 89 proteins (53 and 56 in FL0 and FL+, respectively) with known function and affecting an additional 188 proteins (104 and 126 in FL0 and FL+, respectively) with unknown functions (Source Data). Moreover, the 24 strains showed considerable variability in the ratios of non-synonymous to synonymous mutations (Supplementary Fig. 1b, Source Data).

Almost two-thirds of the mutations were classified as 'low-frequency' (≤10% of the reads), while about one-quarter of the alleles were fully segregated (100% frequency) (Supplementary Fig. 1c, Source Data). In our previous ALE for HL tolerance, we also adapted a batch culture to HL without exposing it to external mutagens in order to increase the mutation rate[35]. This provides a useful point of comparison with our FL-ALE experiment. Comparing the segregation patterns of the mutations obtained during this HL-ALE experiment reveals that the high proportion of low-frequency alleles is likely to be a characteristic of the FL-ALE rather than being due to the absence of external mutagens.

### FL adaptive haplotype

101 mutations were fully segregated in the evolved strains, LT, or WT, but absent from the published *Synechocystis* reference genome (Source Data). Their phylogeny exhibited shorter genetic distances among themselves in the FL0 strains compared to FL+ clones (Fig. 2, Source Data). Of the 101 fully segregated mutations, 44 were not present in LT or WT, including 16 mutations in non-coding regions, 24 protein-altering mutations, and four mutations in structural RNAs. Of these 28 fully segregated mutations, affecting proteins or structural RNAs, five were common to both FL0 and FL+ strains, three were specific to FL0 strains, and 20 were specific to FL+ strains (Table 2, Fig. 2). Three mutations (in *rpoDI*, *sll0518*, *pam68*) occurred in all 24 strains, and five loci harboured multiple mutations. Functionally, the mutated genes were involved in various processes (Table 2).

Three mutations were selected for further analysis: *sll0518*$_{A133V}$, affecting a cyanobacteria-specific protein of unknown function,

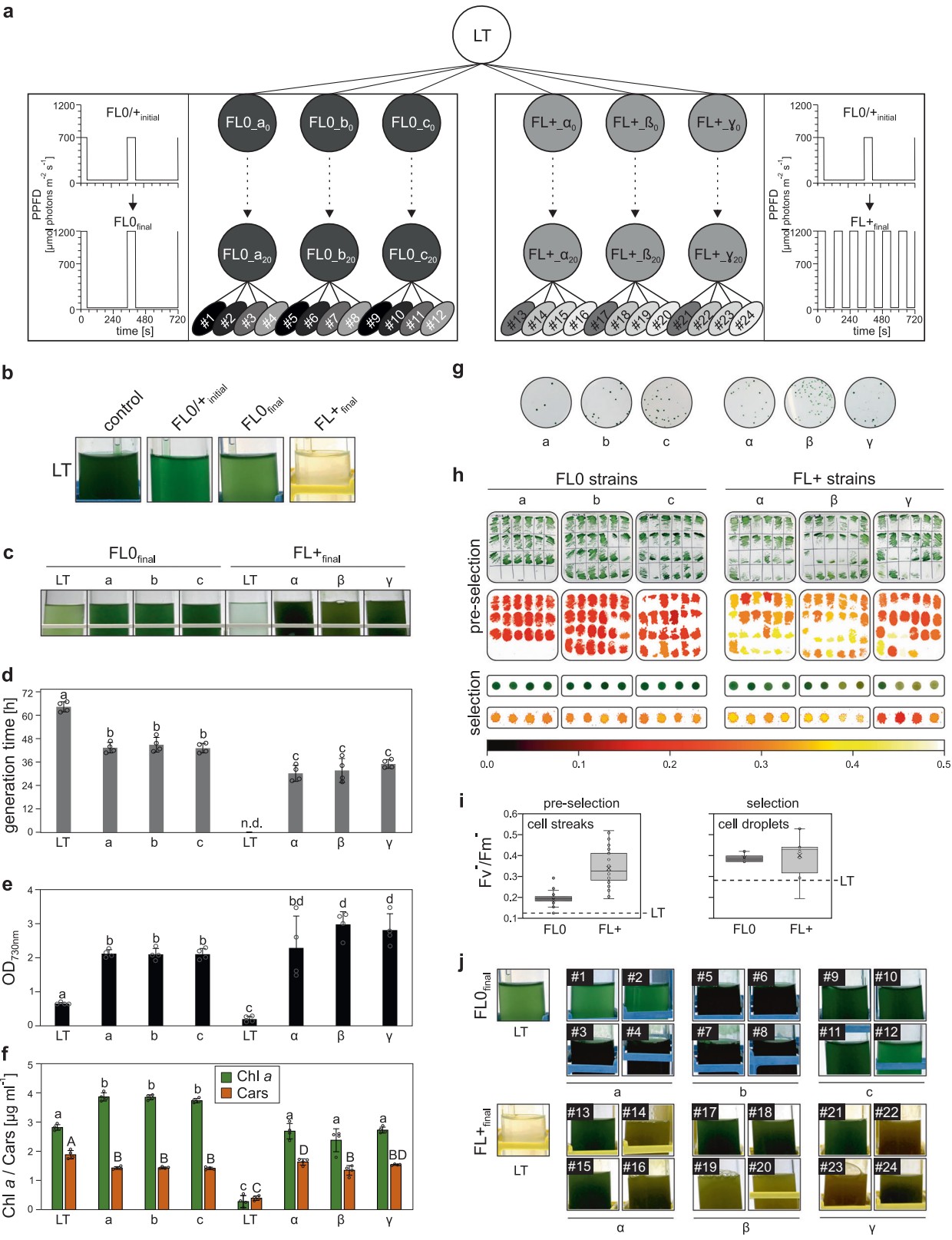

*pam68*<sub></sub> — *pam68_{S113G}*, a missense mutation in *pam68/sll0933*, encoding a factor involved in the early assembly of PSII[40], and *rpaB_{T183P}*, affecting the regulator of phycobilisome association B, the inactivation of which reduces the efficiency of energy transfer from phycobilisomes to PSII[41]. The *sll0518_{A133V}* and *pam68_{S113G}* mutations were common to all 24 strains, suggesting that they arose early in the adaptive process. RpaB_{T183P} was exclusive to three FL+ strains and another mutation of

RpaB (RpaB_{D194G}) independently occurred in three other FL+ strains (Table 2, Fig. 2).

## Recapitulating FL tolerance in LT cells and interplay of FL and HL tolerance

To assess their individual contributions to FL tolerance, the *sll0518_{A133V}*, *pam68_{S113G}*, and *rpaB_{T183P}* mutations were introduced into

**Fig. 1 | Experimental design of the two FL-ALE experiments and the strains generated. a** ALE scheme. Batch cultures (circles), propagation events (dots), and monoclonal cultures (ovals) are shown. *Synechocystis* laboratory type (LT) initiated six independent cultures (three per FL protocol). From each of the six cultures, four monoclonal isolates were selected for whole-genome sequencing and analysis (#1–#24). **b** FL tolerance of LT. LT was grown for 14 days under $LL_{50}$ (control), initial FL (1 min $HL_{700}$/5 min $LL_{50}$), $FL0_{final}$ (1 min $HL_{1200}$/5 min $LL_{12}$) and FL+ (1 min $HL_{1200}$/1 min $LL_{12}$). **c–f** Characterization of adapted cultures. After 7 days under final FL regimes, visual appearance (**c**), growth kinetics (**d**), cell density (**e**), and pigment content (chlorophyll a, carotenoids) (**f**) were measured. Data in (**d**–**f**) are mean ± SD of $n = 4$ biological replicates. Lowercase/Uppercase letters indicate statistically significant differences ($p \leq 0.05$) as determined by two-sided one-way ANOVA with post hoc Bonferroni–Holm corrected Tukey HSD tests. **g** Monoclonal isolation was performed by serial dilution, followed by plating on BG11 agar and incubation

under constant $LL_{20}$. **h** Individual clones were screened in two stages. *Pre-selection*: Clones were grown on agar for 7 days ($LL_{20}$, 23 °C), then assessed for colony morphology and chlorophyll fluorescence (Fv⁻/Fm⁻) across 18–24 clones from panel (**g**). *Selection*: Four clones per set, spanning the Fv⁻/Fm⁻ range, were grown in liquid culture, normalized to $OD_{730} = 10$, and 10 μL spotted on agar for further growth and Fv⁻/Fm⁻ analysis. Colour coding of Fv⁻/Fm⁻ values is shown at the bottom of panel (**h**). **i** Statistical analysis of Fv⁻/Fm⁻ values of pre-selected ($n = 66$ for FL0, 72 for FL+) and selected ($n = 12$ each) clones was performed. Dotted lines indicate Fv⁻/Fm⁻ values for LT control. **j** The 24 selected clones (12 FL0, #1 to #12; 12 FL+, #13 to #24) were cultivated for 14 days under their respective selective final light regimes, with LT controls for comparison. Box plots show individual data points, median (horizontal lines), mean (crosses), interquartile range (box), and 1.5× interquartile range (whiskers). Original data are provided in Source Data.

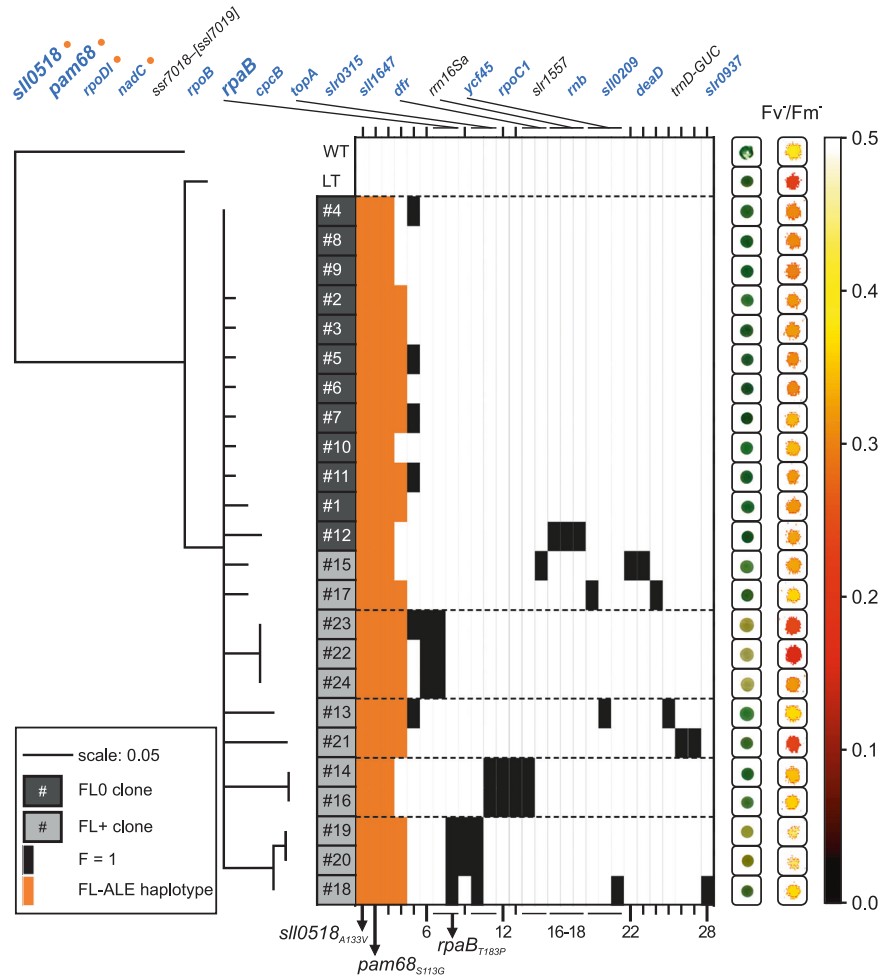

**Fig. 2 | Phylogenetic and mutational analysis of the monoclonal FL-ALE strains.** The left panel displays a maximum likelihood phylogram illustrating the genetic relationship among FL0-ALE (dark grey boxes) and FL+-ALE (light grey boxes) monoclonal strains. The strain numbers (#) correspond to the ones in Fig. 1j. The phylogram includes the original motile, glucose-sensitive *Synechocystis* sp. PCC 6803 isolate (WT) and the laboratory type (LT, non-motile, glucose-tolerant) as reference points. Genetic distances are represented by horizontal solid lines, with a scale of 0.05 substitutions per site. The phylogenetic analysis is based on 101 fully segregated alleles (allele frequency = 1), with only branches showing ≥80% bootstrap support displayed. The evolved strains are grouped into five distinct clades, demarcated by horizontal dashed lines. The central panel depicts a map of 28 mutations absent in WT and LT strains. This includes 24 protein-altering mutations (20 non-synonymous base substitutions and 4 indels) and 4 mutations affecting

rRNAs or tRNAs (refer to Table 1 for details). Mutations present at 100% frequency in at least one monoclonal strain are represented by black or orange rectangles, with orange indicating alleles common to ≥16 of the 24 strains. Loci are arranged left to right based on mutation frequency across the 24 strains, except for mutations affecting the same locus, which are grouped together. Genes affected by non-synonymous SNPs are highlighted in blue. Gene identifiers separated by "/" denote deletions in intergenic regions, while "−" indicates deletions affecting coding regions (affected genes in brackets). Three key alleles - $sll0518_{A133V}$, $pam68_{S113G}$, and $rpaB_{T183P}$ - are emphasized in bold and marked by arrow-heads, respectively. The right panel presents photographic images of representative cell droplets for each monoclonal strain, accompanied by their respective Fv⁻/Fm⁻ values, providing a visual and quantitative phenotypic characterization of the evolved strains.

**Table 2 | Fully segregated, protein-affecting mutations identified in FL-ALE strains**

| # | Locus | Annotation | Mutation | No. | ALE | Function |
|---|-------|-----------|----------|-----|-----|----------|
| 1 | *sll0518* | Unknown protein | **A133V (GCC → GTC)\*** | **24** | **O+** | **?** |
| 2 | *pam68 (sll0933)* | PAM68 protein | **S113G (AGC → GGC)\*** | **24** | **O+** | **P** |
| 3 | *rpoDI (slr0653)* | RNA polymerase σ factor | R96L (CGT → CTT)\* | 24 | O+ | Tc |
| 4 | *nadC (slr0936)* | Nicotinatenucleotide pyrophosphorylase | Q183H (CAA → CAT)\* | 16 | O+ | M |
| 5 | *ssr7018–[ssl7019]* | Hypothetical proteins | Δ935 bp | 6 | O+ | ? |
| 6 | *rpoB (sll1787)* | RNA polymerase beta subunit | T791N (ACC → AAC) | 3 | + | Tc |
| 7 | *rpaB (slr0947)* | OmpR subfamily | **T183P (ACC → CCC)** | **3** | **+** | **Tc** |
| 8 | | | D194G (GAC → GGC) | 3 | + | |
| 9 | *cpcB (sll1577)* | Phycocyanin b subunit | S40P (TCT → CCT) | 2 | + | P |
| 10 | *topA (slr2058)* | DNA topoisomerase I | T504A (ACC → GCC) | 3 | + | Tc |
| 11 | | | E149D (GAA → GAT) | 2 | + | |
| 12 | *slr0315* | Hypothetical protein | V62L (GTT → CTT) | 2 | + | ? |
| 13 | *sll1647* | Hypothetical protein | W97R (TGG → AGG) | 2 | + | ? |
| 14 | *dfr (sll0698)* | Drug sensory protein A | T393A (ACC → GCC) | 2 | + | M |
| 15 | | | F605L (TTC → TTG) | 1 | + | |
| 16 | *rrn16Sa* | 16S ribosomal RNA | noncoding (254/1489 nt) | 1 | O | Tl |
| 17 | | | noncoding (247/1489 nt) | 1 | O | |
| 18 | | | noncoding (243/1489 nt) | 1 | O | |
| 19 | *ycf45 (slr0692)* | Hypothetical protein | Δ13 bp, coding (331-343/1770 nt) | 1 | + | ? |
| 20 | | | E533G (GAA → GGA) | 1 | + | |
| 21 | | | +T, coding (59/1770 nt) | 1 | + | |
| 22 | *rpoC1 (slr1265)* | RNA polymerase γ subunit | R221L (CGG → CTG) | 1 | + | Tc |
| 23 | *slr1557* | Unknown protein | Δ8 bp, coding (573-580/1110 nt) | 1 | + | ? |
| 24 | *rnb (sll1290)* | Ribonuclease II | V500M (GTG → ATG) | 1 | + | Tc |
| 25 | *sll0209* | Hypothetical protein | Q290E (CAA → GAA) | 1 | + | ? |
| 26 | *deaD (slr0083)* | ATP dependent RNA helicase; DeaD | R399L (CGG → CTG) | 1 | + | Tc |
| 27 | *trnD-GUC* | tRNA for aspartate | A → G, noncoding (8/74 nt) | 1 | + | Tl |
| 28 | *slr0937* | Unknown protein | G292S (GGT → AGT) | 1 | + | ? |

The table lists non-synonymous single nucleotide polymorphisms (SNPs) and deletions within coding regions that are absent in both the laboratory type (LT) and wild-type (WT) strains, and have achieved 100% allele frequency in at least one of the 24 monoclonal strains. Columns are organized as follows: "#": Corresponds to the mutation numbering in Fig. 2. "Locus": Gene identifier; "–" denotes deletions within coding regions, with affected genes in brackets "[]". "Mutation": Specifies the nature and position of SNPs or indels, using the notation "x/y nt" where x is the affected position in a sequence of total length y. "No.": Indicates the number of monoclonal strains harbouring the specific allele. "ALE": Specifies the ALE protocol(s) in which the allele was detected (0 for FL0, + for FL +, O+ for both). "Function": Assigns Gene Ontology (GO) terms - M (metabolic processes), P (photosynthesis), Tc (transcription), Tl (translation), and ? (unknown function). Mutations associated with the FL-tolerant haplotype (present in ≥65% of monoclonal strains) are highlighted with an asterisk. Alleles reconstituted and assessed for FL tolerance in this study are highlighted in bold font.

LT at the corresponding wild-type gene loci via marker-less homologous recombination. This generated three sets of strains that differed from LT only in terms of these single SNPs. According to Alphafold3[42], all three amino-acid substitutions were predicted to cause no folding differences in their corresponding proteins (Supplementary Fig. 2a). The $sll0518_{A133V}$ and $pam68_{S113G}$ mutants showed significantly enhanced $FL0_{final}$ tolerance compared to LT (Fig. 3, Source Data). However, these strains failed to grow productively under $FL+_{final}$. In contrast, $rpaB_{T183P}$ displayed growth comparable to LT under $FL0_{final}$ but outperformed all other strains under $FL+_{final}$. Under constant $LL_{12}$, $sll0518_{A133V}$ exhibited slightly increased growth, $rpaB_{T183P}$ showed slightly decreased growth compared to LT, while $pam68_{S113G}$ grew similarly to LT (Supplementary Fig. 2, Source Data). Under constant $HL_{1200}$, lethal for non-adapted LT cells, only $rpaB_{T183P}$ demonstrated productive growth.

To test whether mutations conferring HL tolerance could also impart resistance to $FL+_{final}$ conditions, we evaluated the growth of three previously characterized HL-tolerant strains[35]. However, none of the HL-tolerant strains exhibited growth under $FL+_{final}$ (Supplementary Fig. 3, Source Data), implying that the mechanisms underlying tolerance to constant HL and fluctuating HL may be distinct, and that adaptations to one condition do not necessarily confer tolerance to the other.

In summary, $sll0518_{A133V}$ and $pam68_{S113G}$, identified in all strains adapted to $FL0_{final}$ or $FL+_{final}$, confer enhanced tolerance to $FL0_{final}$ but not to $FL+_{final}$. Conversely, $rpaB_{T183P}$, specific to $FL+_{final}$ adaptation, imparts tolerance to both $FL+_{final}$ and constant HL, even in the absence of the four common mutations found in the FL-adapted haplotypes. This suggests that the FL-adapted haplotype mutations are not prerequisite for $FL+_{final}$ or HL tolerance and that mutations conferring HL tolerance do not necessarily provide tolerance to $FL+_{final}$.

## $Pam68_{S113G}$ enhances the accumulation and activity of PSII under FL

To understand the effects of the $pam68_{S113G}$ mutation, we created a Pam68 overexpression strain ($pam68oe$) and compared its growth to a $pam68$ knockout mutant ($insO933$)[40] under both $LL_{12}$ and $FL0_{final}$ (Supplementary Fig. 4a–c, Source Data). Under $FL0_{final}$, $insO933$ exhibited slower growth than LT for up to 120 h. This was followed by a sudden acceleration, resulting in an $OD_{730nm}$ seven days after inoculation that was 32% higher than that of LT. The final $OD_{730nm}$ of $pam68_{S113G}$, in contrast, was 92% higher. The $pam68oe$ strain grew 67% better than LT, slightly less than $pam68_{S113G}$, but not significantly different from $insO933$ and $pam68_{S113G}$. No significant growth differences were seen among the strains under $LL_{12}$.

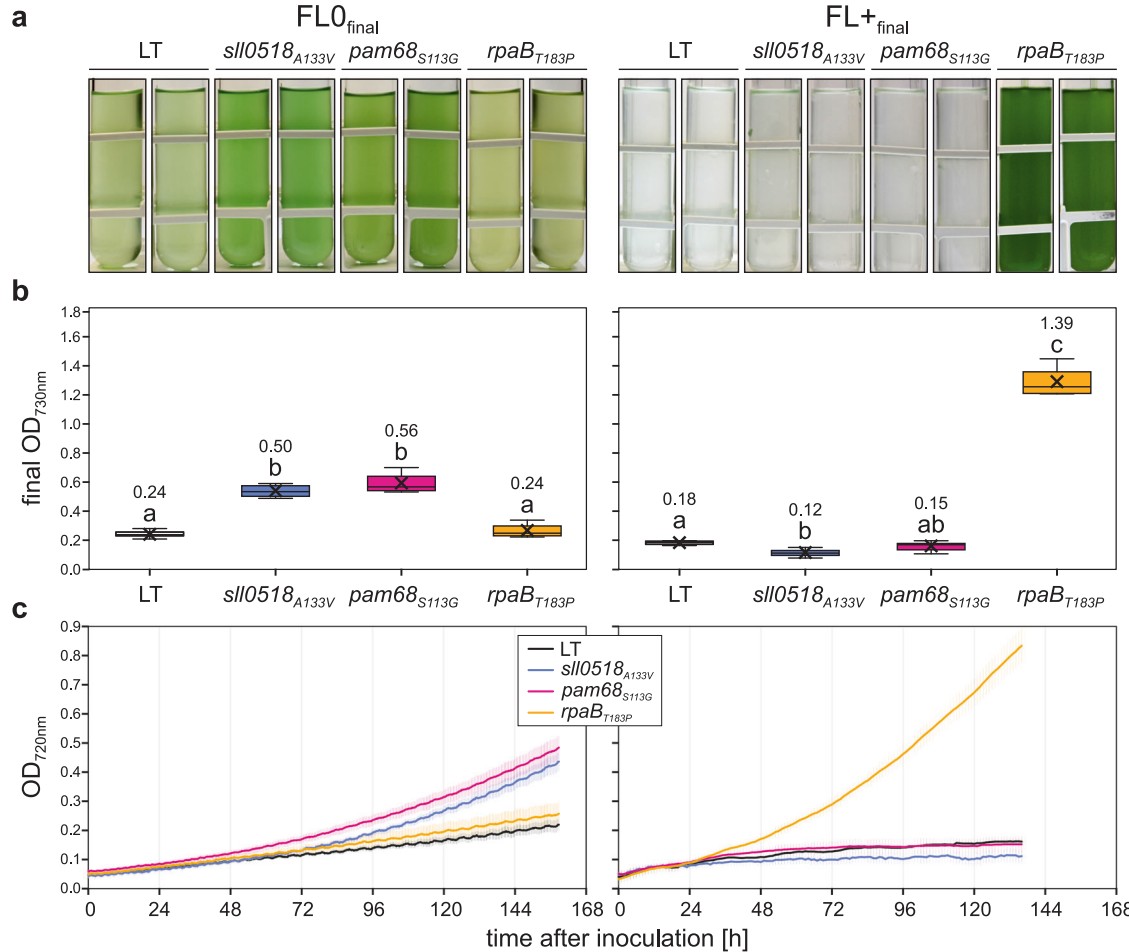

**Fig. 3 | Growth characteristics of *sll0518_{A133V}*, *pam68_{S113G}*, *rpaB_{T183P}* and LT strains under FL0_{final} and FL+_{final} conditions. a** Visual representation of liquid cultures for the four strains cultivated in multi-cultivators under FL0_{final} (left) or FL +_{final} (right) conditions at 23 °C with 100 mL min⁻¹ aeration, photographed seven days post-inoculation. **b** Quantitative analysis of cell density (OD_{730nm}) for the four strains grown under conditions described in a. Box plots are presented, with lowercase letters denoting statistically significant differences ($p \leq 0.05$) as determined by post-hoc Bonferroni–Holm simultaneous comparison of all measurements ($n = 8$ biological replicates) following significant between-group differences detected by one-factorial ANOVA. **c** Growth kinetics of the four strains under FL0_{final} and FL+_{final} conditions, monitored automatically by multi-cultivators measuring OD_{720nm}. The solid lines show the mean, and error bars represent the standard deviation ($n = 4$ biological replicates, except for sll0518_{A133V} and pam68_{S113G} under FL+, where $n = 8$ biological replicates). Statistical data in panel (**b**) are visualized using box plots, showing individual data points, median (horizontal lines), mean (crosses), interquartile range (box), and 1.5× interquartile range (whiskers). Raw data are available in Source Data.

Since Pam68 is involved in PSII assembly[40,43], we examined thylakoid protein levels. To this end, we grew LT and two independent *pam68_{S113G}* mutant strains under FL0_{final} conditions. We also tested our standard LL control condition (LL_{50}) and found that *pam68_{S113G}* cultures had a significantly higher OD_{730nm} than LT under this condition (Fig. 4a–c, Source Data). Therefore, we used the LL_{50} and FL0_{final} conditions to study thylakoid protein accumulation in the *pam68_{S113G}* and LT strains. Immunoblot analysis was used to quantify the levels of Pam68 and representative thylakoid proteins (D1, PsaA, AtpB), while allophycocyanin (APC) and phycocyanin (PC) were quantified via in-gel fluorescence (Fig. 5a, b; Source Data). Under LL_{50} conditions, the levels of Pam68 (+1%), D1 (+7%), PsaA (−9%), and AtpB (+8%) were found to be non-significantly altered in *pam68_{S113G}* compared to LT, while APC + PC levels were found to be significantly increased (+14%; $p = 1.94 \times 10^{-3}$). Under FL0_{final} conditions, *pam68_{S113G}* cells showed reduced levels of Pam68 (−17%; $p = 4.90 \times 10^{-2}$) and APC + PC (−9%; $p = 3.88 \times 10^{-4}$) compared to LT. Meanwhile, the levels of D1 (+35%; $p = 3.45 \times 10^{-5}$) and PsaA (+16%; $p = 1.14 \times 10^{-3}$) were increased, while AtpB levels remained unchanged. These results suggest an increase in photosynthetic complex accumulation in *pam68_{S113G}* mutants under FL0_{final} conditions, but not under constant LL_{50} conditions. This also suggests that

*pam68_{S113G}* is likely a gain-of-function mutation, as its beneficial effect on growth under FL0_{final} exceeds that of Pam68 overexpression, despite lowered Pam68 protein levels. The reasons for the slight decrease in mutant Pam68 levels compared to WT Pam68 under FL conditions remain unclear.

The serine that mutated to glycine at position 113 in *Synechocystis* Pam68 is conserved from cyanobacteria to flowering plants, corresponding to serine at position 174 of *Arabidopsis thaliana* PAM68 (AtPAM68, At4g19100)[40]. To evaluate the adaptive potential of this S → G mutation across species, WT *AtPAM68* (*AtPAM68_{WT}*) and *AtPAM68_{S174G}* were expressed in *Synechocystis* LT and *ins0933*, and growth was observed under LL_{50}, HL_{700} and FL0_{final} conditions (Fig. 4a–c). No increase in growth was observed under LL_{50} and FL0_{final} conditions compared to the corresponding controls (LT or *ins0933*). However, the overexpression of *AtPAM68_{WT}* in the *ins0933* mutant background significantly increased growth at HL_{700}. Conversely, the mutated *AtPAM68_{S174G}* did not exhibit this effect in the *ins0933* background. This demonstrates that, despite the evolutionary distance between land plants and cyanobacteria, the plant PAM68 can still functionally replace its cyanobacterial counterpart. Moreover, the importance of this amino acid position is also

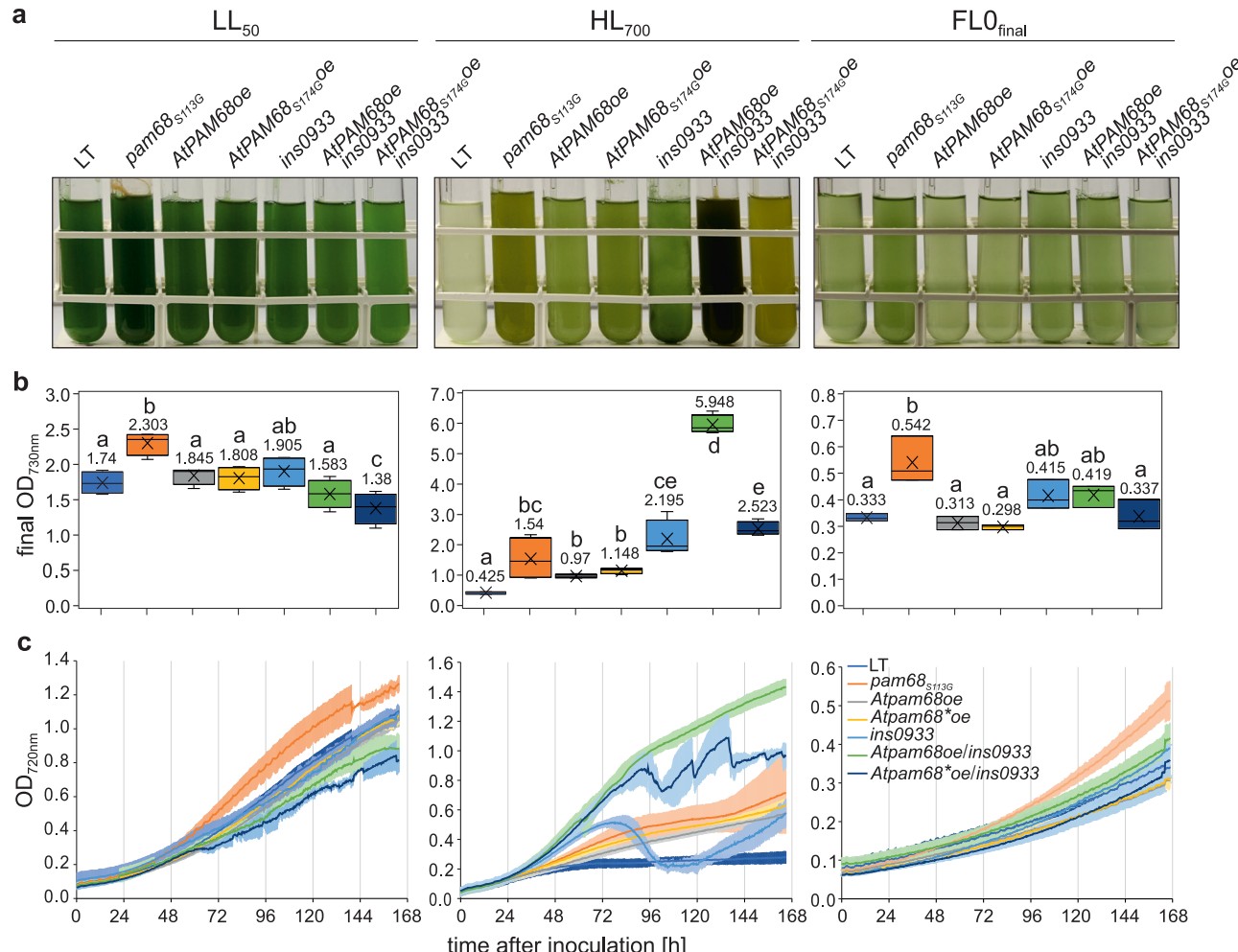

**Fig. 4 | Growth characteristics of *pam68*$_{S113G}$, *ins0933*, *AtPAM68oe* and *AtPAM68*$_{S174G}$*oe* strains and LT strains under three different light conditions. a** Visual representation of liquid cultures of the LT, *pam68*$_{S113G}$, *ins0933*, *AtPAM68oe* and *AtPAM68*$_{S174G}$*oe* strains cultivated in multi-cultivators under LL$_{50}$, HL$_{700}$, and FLO$_{final}$ conditions at 23 °C with 100 mL min$^{-1}$ aeration, photographed seven days post-inoculation. For the overexpression of plant PAM68 protein, the codon-optimized *PAM68* CDS sequence from *Arabidopsis thaliana* (AT4G19100.1), without the chloroplast transit peptide (cTP), was used. **b** Quantitative analysis of cell density

(OD$_{730nm}$) for the strains grown under the conditions as in (**a**). Box plots are presented as in Fig. 3b (*n* = 4 biological replicates for LL$_{50}$ and HL$_{700}$, *n* = 3 biological replicates for FLO$_{final}$). **c** Growth kinetics of the strains grown under conditions as in (**b**), monitored automatically by multi-cultivators measuring OD$_{720nm}$. The solid lines show the mean, and error bars represent the standard deviation. Statistical data in panel (**b**) are presented as box plots, showing individual data points, median (horizontal lines), mean (crosses), interquartile range (box), and 1.5× interquartile range (whiskers). Raw data are provided in Source Data.

conserved in the two Pam68 proteins, with a serine-to-glycine exchange resulting in pronounced phenotypic changes under specific light conditions. However, it seems that the rest of the protein sequence determines whether an increased tolerance to FLO$_{final}$ or HL$_{700}$ results from a serine or a glycine at this conserved position (glycine in *Synechocystis* Pam68 for tolerance to both FLO$_{final}$ and HL$_{700}$; serine in AtPAM68 for HL$_{700}$ tolerance). Therefore, it can be concluded that the role of PAM68 in FL tolerance is most likely not conserved in flowering plants.

To investigate the role of Pam68$_{S113G}$ in the increase of PSII levels (see Fig. 5a, b), we analysed PSII complex assembly using two-dimensional clear-native (CN)/SDS-PAGE, followed by immunoblot analysis (Fig. 5c, Source Data). In-gel Chl *a* fluorescence indicated an increase in PSII dimer abundance under both LL$_{50}$ and FLO$_{final}$ conditions (Fig. 5c), thus corroborating the results of the SDS-PAGE immunoblots. Analysis of the second dimension by immunoblotting showed a similar distribution of Pam68 and Pam68$_{S113G}$ signals between the low- and high-molecular-weight fractions, indicating no overall change in the interaction patterns of mutant Pam68 and corroborating the

SDS-PAGE immunoblot results (see Fig. 5a, b). Furthermore, pulse labelling experiments showed that the *pam68*$_{S113G}$ mutant exhibited a pronounced reduction in *de-novo* biosynthesis of membrane proteins, with a clear decline in unassembled D1 and CP43 protein in RCIIa and CP43m assembly intermediates as compared to LT under LL$_{50}$ (Supplementary Fig. 5). This reduction was confirmed by immunodetection of D1 and CP43 on the 2D blot (Fig. 5c), which may indicate enhanced stability of mature PSII complexes, as these showed no decrease in steady-state levels (Fig. 5a, b)[44].

To further assess the physiological effects of the S113G mutation of Pam68, PSII activity, respiration rates, apparent PSII quantum yield (Fv'/Fm') and P700 oxidation kinetics were determined (Supplementary Fig. 6). Under LL$_{50}$, PSII activity, as measured as O$_2$ evolution (see Methods), was slightly lower in *pam68*$_{S113G}$ than in LT when normalized to OD$_{730nm}$, but slightly higher when normalized to Chl *a* (Supplementary Fig. 6a). This suggests that the *pam68*$_{S113G}$ mutant has a lower Chl *a*/OD$_{730nm}$ ratio. Under FLO$_{final}$, however, PSII activity significantly increased per unit OD$_{730nm}$ (+227%; *p* = 1.22 × 10$^{-23}$) and per mg Chl *a* (+212%; *p* = 5.11 × 10$^{-22}$). Respiration per unit OD$_{730}$ was moderately

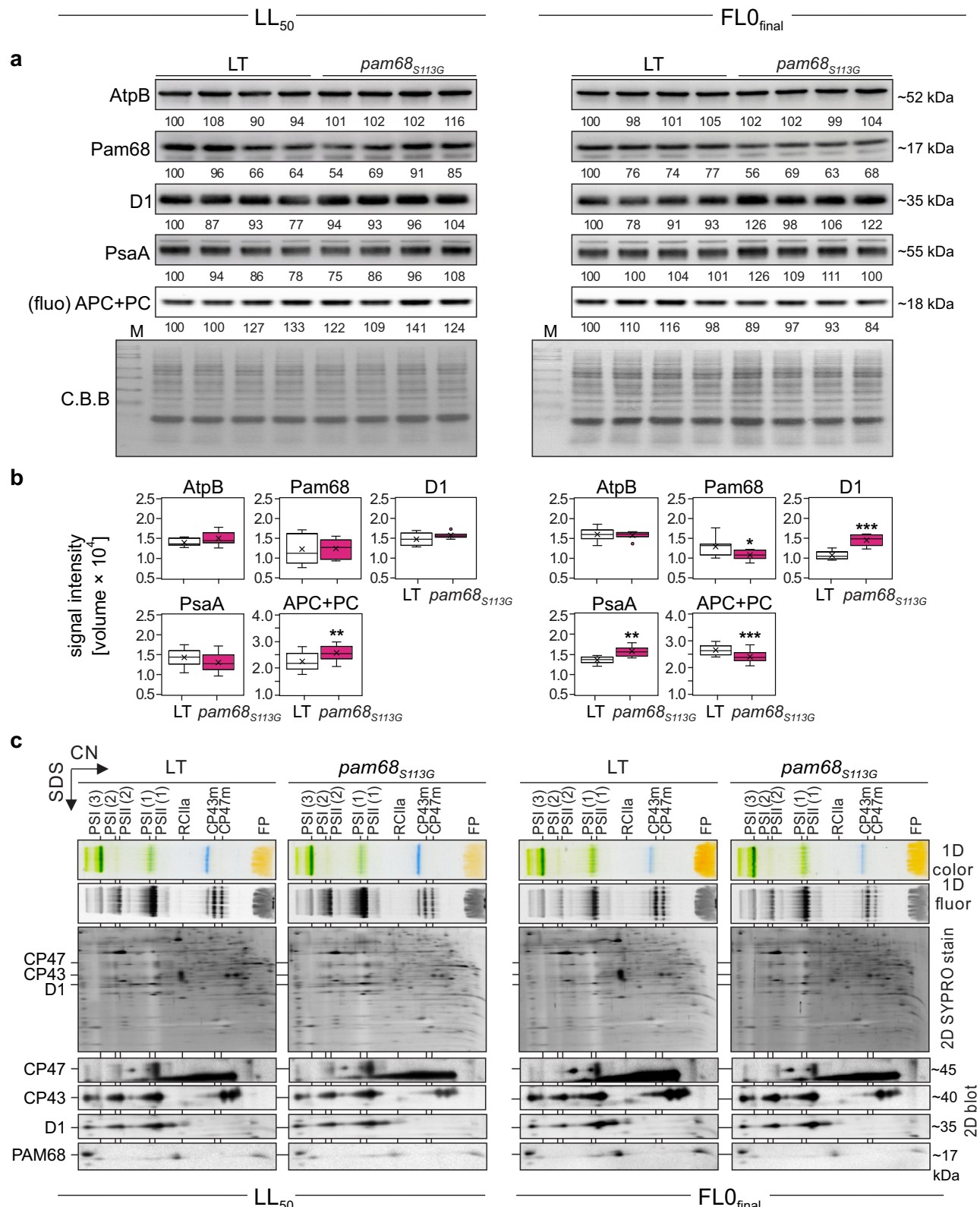

decreased ($-16\%$; $p = 2.70 \times 10^{-2}$) under $LL_{50}$ and not significantly decreased ($-5\%$; $p = 7.21 \times 10^{-1}$) under $FL0_{final}$ (Supplementary Fig. 6b). In-vivo fluorimetry (FluorCam) also indicated an increase in $Fv^-/Fm^-$ in $pam68_{S113G}$ mutants adapted to $LL_{20}$ ($+21\%$; $p = 4.55 \times 10^{-16}$) and $FL0_{final}$ ($+118\%$; $p = 2.75 \times 10^{-21}$) (Supplementary Fig. 6c), further suggesting that $pam68_{S113G}$ enhances PSII performance. A similar trend was observed using a DualPam fluorometer (Source Data)

In addition, $pam68_{S113G}$ was also observed to display delayed P700 oxidation when exposed to far-red (FR) light following a prolonged period of incubation in the dark, and to undergo accelerated P700 re-reduction when FR was switched off (Supplementary Fig. 6d, e). This corresponds to an increase in the time required for half-maximum P700 oxidation ($t_{0.5}P700_{ox}$) under $LL_{50}$ ($+50\%$; $p = 6.3 \times 10^{-10}$) and $FL0_{final}$ ($+15\%$; $p = 5.20 \times 10^{-4}$) conditions, and a decrease in the half-

**Fig. 5 | The $pam68_{S113G}$ mutation enhances photosystem accumulation at FL.**
**a** Immunoblot analysis. Whole-cell protein extracts were obtained and aliquots containing ~0.35 µg of Chl a content were subjected to SDS-PAGE. Chromophore fluorescence from phycobilisome marker proteins (allophycocyanin and phycocyanin, APC + PC) was visualized directly from the gels. Proteins were electro-transferred onto PVDF membranes. Specific antibodies were employed for immunodetection of Pam68, D1, PsaA and AtpB. A Coomassie Brilliant Blue (C.B.B) stain of the PVDF membrane served as a loading control. Relative quantification of immunoblot signals was performed, with values normalized to the first lane of each blot. Comprehensive data analysis is presented in panel (**c**) and Source Data. Molecular weight markers are indicated. **b** Quantification of signals for Pam68, D1, PsaA, AtpB, and APC + PC was performed using 4 biological replicates, each analysed in duplicate ($n = 8$), except for APC + PC, which was assessed using 4 biological replicates with five technical replicates each ($n = 20$). Data are presented as box plots, showing individual data points, median (horizontal lines), mean (crosses), interquartile range (box), and 1.5× interquartile range (whiskers).

Statistical significance was determined using two-sided Student's *t* Test, with *$p \leq 0.05$, **$p \leq 0.01$, and ***$p \leq 0.001$. $p = 1.41 \times 10^{-1}$ for AtpB, $9.17 \times 10^{-1}$ for Pam68, $1.24 \times 10^{-1}$ for D1, $2.99 \times 10^{-1}$ PsaA, and $1.94 \times 10^{-3}$ for APC + PC under $LL_{50}$, and $7.70 \times 10^{-1}$ for AtpB, $4.90 \times 10^{-2}$ Pam68, $3.43 \times 10^{-5}$ for D1, $1.41 \times 10^{-3}$ PsaA, and $3.88 \times 10^{-4}$ for APC + PC under $FLO_{final}$, respectively. **c** Two-dimensional PAGE (CN/SDS-PAGE) analysis of thylakoid proteins. Thylakoid preparations (equivalent to ~4 µg chlorophyll) were solubilized and separated in two dimensions. The 1D native gels were photographed (1D colour) and examined for Chl fluorescence (1D fluor). The 2D SDS-PAGE gels were electro-transferred onto PVDF membranes and stained with SYPRO. The membranes were then immunoblotted using antibodies raised against D1, CP43, CP47 and Pam68. The assembly complexes are annotated as described[43]: monomer/dimer/trimer (1/2/3), reaction centre complex lacking PSII core antenna modules CP43 and CP47 (RCIIa), and CP43/47 module (CP43m/47 m). FP: free pigments. Experiments were repeated independently two times with similar results, and representative images of one biological replicate are shown. Raw data are provided in Source Data.

time of re-reduction ($t_{0.5}P700_{red}$) ($LL_{50}$: −66%; $p = 2.33 \times 10^{-12}$; $FLO_{final}$: −43%; $p = 2.46 \times 10^{-7}$) (Supplementary Fig. 6d, e). High $t_{0.5}P700_{ox}$ and low $t_{0.5}P700_{red}$ values can be taken as an indirect measure of high CEF, as the oxidation of P700 upon exposure to FR light following the depletion of respiratory donors by prolonged dark incubation is primarily mediated by CEF[45,46]. Increased CEF may support improved FL tolerance by enhancing ΔpH formation and thereby restricting electron flow through cytochrome $b_6f$ into PSI (photosynthetic control) while temporarily alleviating PSI acceptor site limitation upon HL onset during FL cycles. However, under $LL_{50}$ conditions, $pam68_{S113G}$ likely contains less PSI than LT, as suggested by its ~20% lower Chl a content (Source Data). Consequently, a relative increase of respiratory electron input via cytochrome $b_6f$ may also contribute to the slower P700 oxidation and faster re-reduction relative to LT in $LL_{50}$ samples. Consistent with this, chlorophyll content does not differ significantly between genotypes under FL0 conditions, and differences in P700 kinetics are correspondingly less pronounced.

### The $rpaB_{T183P}$ mutation is associated with downregulation of light harvesting

RpaB/Slr0947 plays a crucial role in regulating energy transfer from phycobilisomes to photosystems[47–49]. This suggests that reduced light harvesting capacity may contribute to the enhanced tolerance of $rpaB_{T183P}$ cultures to HL and FL+$_{final}$ that are lethal to the starter strain. To investigate the functional implications of this mutation, an RpaB overexpression strain ($rpaBoe$) was generated and compared with a previously characterized RpaB knock-down mutant ($rpaBkd$)[49] under constant $HL_{1200}$ and FL+$_{final}$. Notably, only the $rpaB_{T183P}$ mutant demonstrated productive growth under both conditions, while LT, $rpaBkd$, and $rpaBoe$ strains failed to grow (Supplementary Fig. 7, Source Data). Furthermore, LT and $rpaB_{T183P}$ had similar endpoint $OD_{730nm}$ values under $LL_{50}$ (Fig. 6a, Source Data). However, under $HL_{700}$, $rpaB_{T183P}$ showed markedly higher values, and under FL+$_{final}$, only $rpaB_{T183P}$ grew while LT failed.

Immunoblot analysis and in-gel fluorescence quantification showed slightly increased RpaB levels in $rpaB_{T183P}$ under $LL_{50}$ (+37%; $p = 1.10 \times 10^{-3}$) and $HL_{700}$ (+22%; $p = 1.94 \times 10^{-3}$) conditions, compared to LT. At $LL_{50}$, no significant changes were observed in D1 (+11%; $p = 2.30 \times 10^{-1}$), PsaA (+6%; $p = 2.74 \times 10^{-1}$), or AtpB levels (−3%; $p = 5.31 \times 10^{-1}$), while APC + PC levels were significantly reduced (−22%; $1.03 \times 10^{-8}$) in $rpaB_{T183P}$ (Fig. 6b, c; Source Data). Under $HL_{700}$, D1 levels were significantly reduced (−21%; $p = 2.32 \times 10^{-3}$), while non-significant changes in PsaA (+16%), AtpB (+2%) and APC + PC levels (−15%) were observed (Fig. 6b, c). This indicates a general downregulation of peripheral antennas in $rpaB_{T183P}$, with differential effects on PSI and PSII under LL and HL conditions. Consistently, the ratio between PC and Chl a absorption maxima was significantly reduced in $rpaB_{T183P}$ under

both $LL_{50}$ and $HL_{700}$ (Supplementary Fig. 8a, Source Data), indicating downregulation of light harvesting.

A comparison of the 77 K emission spectra of blue-light-treated and dark-treated cells grown in $LL_{50}$ revealed a decrease in the S2-to-S1 state transition in $rpaB_{T183P}$ compared to LT (−9% at an excitation wavelength of 600 nm; $p = 6.85 \times 10^{-5}$) (Fig. 6d). The same trend was observed in cells grown at $HL_{700}$ (−9%, $p = 3.81 \times 10^{-2}$), thus supporting previous suggestions that RpaB regulates state transitions[50]. Importantly, $HL_{700}$-acclimated cells showed strong evidence of blue-light-induced phycobilisome decoupling in both LT and $rpaB_{T183P}$, as indicated by a strong increase in emission around 650 nm. No such effect was observed in $LL_{50}$-acclimated cells (Fig. 6d).

Fluorescence emission spectra were collected at low temperature (77 K) with an excitation wavelength of 435 nm. This revealed a significantly lower PSI:PSII ratio in $LL_{50}$-acclimated $rpaB_{T183P}$ cells compared to LT (−19%; $p = 1.25 \times 10^{-8}$), but a significantly higher ratio in $HL_{700}$-acclimated cells (+30%; $p = 1.12 \times 10^{-5}$) (Supplementary Fig. 8b, c). These results are consistent with the quantification of D1 and PsaA immunoblots (Fig. 6b, c). Moreover, 77 K analyses performed at an excitation wavelength of 600 nm (Source Data) showed a statistically significant increase in the PC:APC ratio in $rpaB_{T183P}$ under $LL_{50}$ (+27%; $p = 3.96 \times 10^{-14}$) and $HL_{700}$ (+43%; $p = 7.99 \times 10^{-16}$) conditions (Supplementary Fig. 8c), which suggests an increase in average rod length under both light conditions. Meanwhile, PSII:APC levels in $rpaB_{T183P}$ significantly increased under both $LL_{50}$ (+16%; $p = 2.56 \times 10^{-7}$) and $HL_{700}$ (+17%; $p = 5.13 \times 10^{-5}$) conditions (Supplementary Fig. 8c). These observations suggest that $rpaB_{T183P}$ has a differential effect on photosystem stoichiometry and peripheral antenna accumulation: PSII accumulation increases under continuous LL, while phycobilisome accumulation decreases under LL and moderate HL. In vivo fluorescence measurements under orange-red actinic light also revealed that non-photochemical quenching (qN)[51] in $rpaB_{T183P}$ cells increased in $LL_{50}$-acclimated cells and decreased in $HL_{700}$-acclimated cells compared to LT (Supplementary Fig. 8d, e). These results suggest that state transitions, the main contributor to qN in cyanobacteria under low light[38,52], may have been increased in $LL_{50}$-acclimated and decreased in $HL_{700}$-acclimated $rpaB_{T183P}$ mutants compared to the LT control. Notably, the increase in qN in $LL_{20}$-acclimated $rpaB_{T183P}$ mutants, despite a decrease in the S2-to-S1 ratio (Fig. 6d), suggests a disproportionate change in other qN components.

Next, we determined the physiological effects of the T183P mutation of RpaB on PSII activity, respiration rate, $Fv^-/Fm^-$, and P700 oxidation kinetics. In addition to $LL_{50}$ conditions, only the effect of $HL_{700}$ could be investigated, as experiments with FL+$_{final}$ conditions were impossible (it was lethal to the LT control), and $FLO_{final}$ conditions were not informative ($rpaB_{T183P}$ mutants performed LT-like under this condition; see Fig. 3). Moreover, P700 oxidation could not be

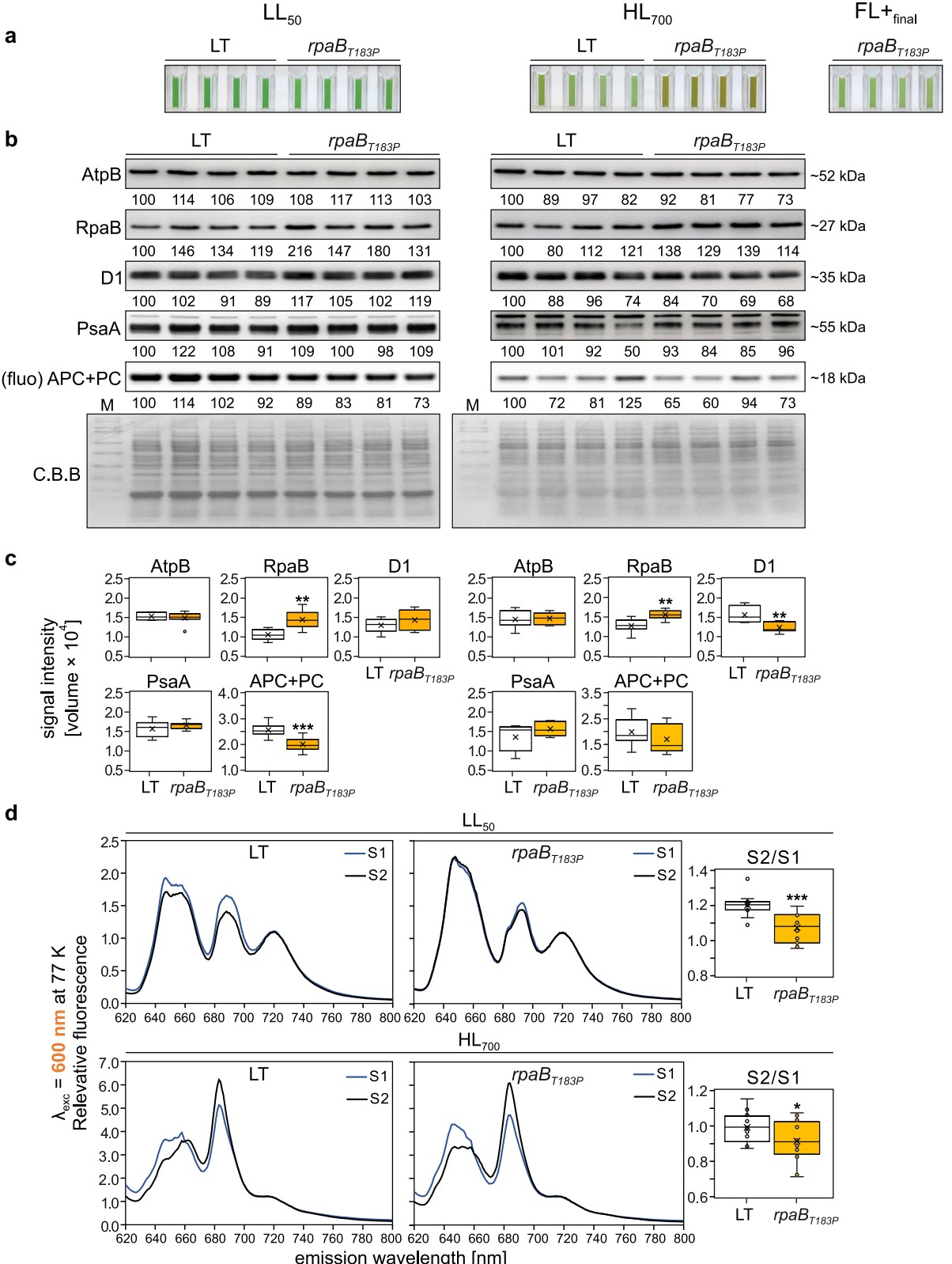

monitored under $HL_{700}$ conditions due to P700 overreduction in both genotypes. Under $LL_{50}$ conditions, PSII activity increased significantly in the presence of DCBQ in $rpaB_{T183P}$ mutants, both per unit $OD_{730nm}$ (+35%; $p = 1.54 \times 10^{-5}$) and per mg Chl $a$ (+35%; $p = 3.68 \times 10^{-5}$), compared to LT (Supplementary Fig. 9a). The respiration of $rpaB_{T183P}$ did not change significantly under $LL_{50}$ compared to LT (−3%; $p = 7.76 \times 10^{-1}$), but increased significantly under $HL_{700}$ per unit $OD_{730}$

(+107%; $p = 6.31 \times 10^{-5}$) (Supplementary Fig. 9b). Consistently, $Fv^{-}/Fm^{-}$ of $rpaB_{T183P}$ cells incubated at $LL_{20}$ was significantly higher than that of LT cells (+72%; $p = 2.45 \times 10^{-38}$) (Supplementary Fig. 9c, Source Data). At the same time, $rpaB_{T183P}$ showed a P700 oxidation and re-reduction behaviour indicative of a substantial rise in CEF activity. This is evidenced by a notably extended P700 oxidation half-time (+54% compared to LT; $p = 2.33 \times 10^{-13}$) and a significantly reduced re-reduction

**Fig. 6 | The _rpaB_<sub>T183P</sub> mutation is associated with downregulation of light harvesting. a** Liquid cultures of LT and _rpaB_<sub>T183P</sub> strains grown under LL$_{50}$, HL$_{700}$ and FL+$_{final}$ (_rpaB_<sub>T183P</sub> only) at 23 °C with aeration were photographed after seven days. **b** Immunoblot analysis was performed on whole-cell extracts. Equal chlorophyll amounts (-0.22 μg for LL$_{50}$, -0.11 μg for HL$_{700}$) were separated by SDS-PAGE. Phycobilisome fluorescence (APC + PC) was recorded directly, and proteins were transferred to PVDF membranes. RpaB, D1, PsaA, and AtpB were detected using specific antibodies, with Coomassie staining as loading control. Signals were quantified relative to the first lane; full analysis is in panel (**c**) and Source Data. **c** Quantification of signals for RpaB, D1, PsaA, AtpB, and APC + PC was carried out using 4 biological replicates, each analysed in duplicate ($n = 8$), except for APC + PC, which was evaluated using 4 biological replicates with five technical replicates each ($n = 20$). $p = 5.31 \times 10^{-1}$ for AtpB, $1.10 \times 10^{-3}$ for RpaB, $2.30 \times 10^{-1}$ for D1, $2.74 \times 10^{-1}$

PsaA, and $1.03 \times 10^{-8}$ for APC + PC under LL$_{50}$, and $7.86 \times 10^{-1}$ for AtpB, $1.94 \times 10^{-3}$ for RpaB, $2.32 \times 10^{-3}$ for D1, $1.38 \times 10^{-1}$ PsaA, and $7.22 \times 10^{-2}$ for APC + PC under HL$_{700}$, respectively. **d** State transitions were analysed in cells grown under LL$_{50}$ and HL$_{700}$. Cells were driven to State 1 (blue light) or State 2 (dark), frozen, and measured at 77 K with 600 nm excitation. Spectra were normalized to PSI emission (F725). Mean spectra are shown ($n = 15$ biological replicates for LL$_{50}$, $n = 16$ for HL$_{700}$). Box plots display S2/S1 ratios derived from F725:F695. $p = 6.85 \times 10^{-5}$ for LL$_{50}$, and $3.81 \times 10^{-2}$ for HL$_{700}$, respectively. Box plots (**c** and **d**) show individual data points, median (horizontal lines), mean (crosses), interquartile range (box), and 1.5× interquartile range (whiskers). Statistical significance was determined using two-sided Student's $t$ Test, with *$p \leq 0.05$, **$p \leq 0.01$, and ***$p \leq 0.001$. Raw data are provided in Source Data.

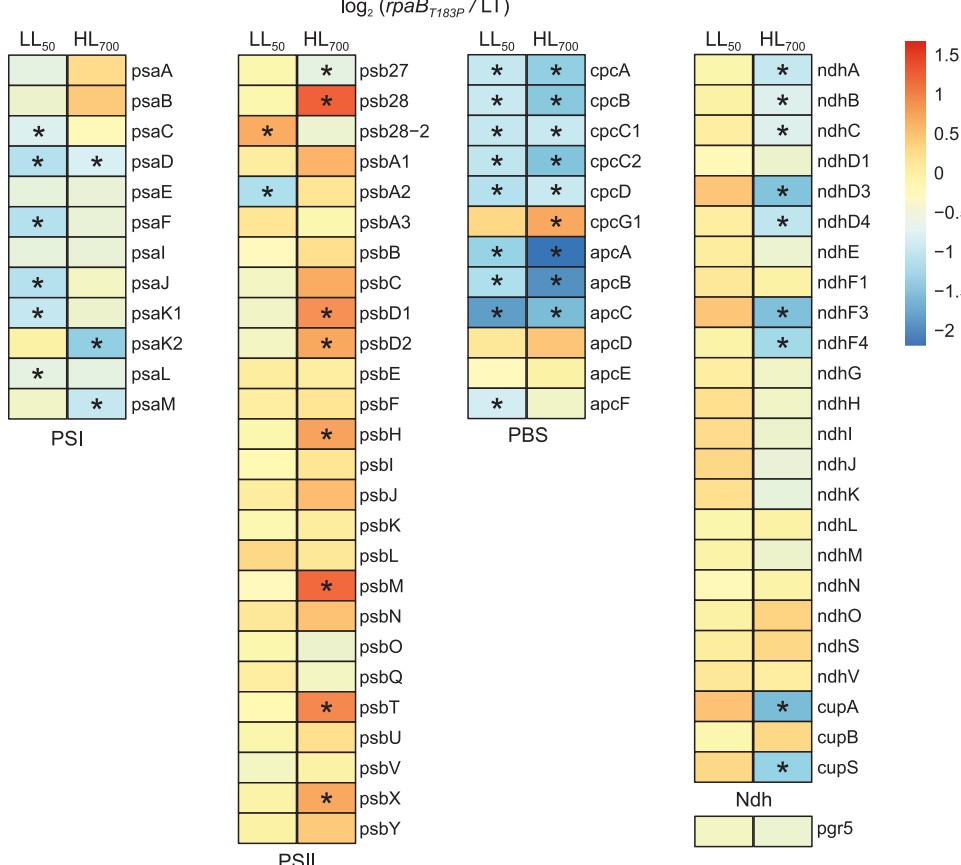

**Fig. 7 | Changes in the transcript levels of photosynthesis-relevant genes in _rpaB_<sub>T183P</sub> compared to LT under LL$_{50}$ and HL$_{700}$ conditions.** The heatmaps illustrate the log$_2$ transformed fold change values of _rpaB_<sub>T183P</sub>/LT for selected genes under LL$_{50}$ and HL$_{700}$, respectively. Down-regulation is represented by blue, and up-regulation by red. Cells highlighted with an asterisk indicate a significant difference (Benjamini–Hochberg adjusted $p < 0.05$). Raw data are provided in Source Data.

half-time (−76% compared to LT; $p = 4.4 \times 10^{-5}$)(Supplementary Fig. 9d, e).

Taken together, these results suggest that the photosynthetic electron transport chain in _rpaB_<sub>T183P</sub> undergoes profound changes, reducing PSII abundance and phycobilisome-mediated light harvesting under moderate HL, while maintaining PSII efficiency under LL conditions.

Consistent with the immunoblot and 77 K analyses, an mRNA sequencing experiment on cells grown under LL$_{50}$ or HL$_{700}$ conditions showed that the RpaB$_{T183P}$ exchange had only a minor effect on the expression of most photosynthetic genes. Under LL$_{50}$ conditions, a uniform downregulation of PSI structural subunit and phycobilisome-related genes was observed (Fig. 7). Conversely, under HL$_{700}$, most PSII

structural subunit genes were found to be upregulated, while many NDH genes, some PSI genes and most phycobilisome-related genes were downregulated in _rpaB_<sub>T183P</sub> (Fig. 7). This contrasts with the results of immunoblot and 77 K analyses, which indicated lowered PSII and increased PSI levels under HL$_{700}$. These findings suggest a complex response involving altered transcript accumulation as a consequence of the T183P substitution. This altered transcript accumulation could be due to an altered binding affinity of the mutated RpaB for its target genes, an altered target spectrum, or secondary changes to the transcriptome in response to the physiological effects of the mutation. Nevertheless, the mutation does not result in corresponding changes in protein levels with respect to the two photosystems, but it does with respect to phycobilisome-related genes.

## Discussion

Fluctuations in light intensity represent one of the most rapid and severe environmental stressors that photosynthetic organisms must cope with[25,53]. Consequently, the molecular mechanisms underlying FL tolerance and their application in crop improvement are being intensively investigated. ALE using cyanobacteria as chloroplast proxies has been previously employed to address HL-related stress[34–36]. In this study, evolutionary screening under two complex FL regimes identified previously unknown FL tolerance factors and adaptive alleles. Among 412 candidate mutations, three non-synonymous SNPs in genes encoding the protein Sll0518 with unknown function, the PSII assembly factor Pam68, and the Response Regulator RpaB, were reconstituted in the parental LT background and confirmed to confer varying yet specific FL adaptation, thus demonstrating that FL tolerance can be improved through ALE.

The $sll0518_{A133V}$ mutation promotes growth under both non-lethal $FLO_{final}$ and $LL_{12}$, but not $HL_{1200}$. Furthermore, it does not facilitate tolerance to $FL+_{final}$ (see Fig. 3 and Supplementary Fig. 2), indicating a specific role in FL acclimation. This mutation was observed in all monoclonal strains (see Table 2, Fig. 2), further supporting its adaptive nature. Sll0518 has recently been found to co-immunoprecipitate with the RNA recognition motif protein Rbp3, which interacts with ribosomes and the 3′-ends of mRNAs encoding photosynthesis proteins and the absence of which lowers the PSI:PSII ratio[54]. The precise function of Sll0518 remains unclear, but it can be speculated that the protein may play an indirect role (via Rpb3) in the accumulation of PSI, which is a key target of FL-induced photodamage[25].

The $pam68_{S113G}$ mutation, present in all 24 monoclonal strains, enhanced $FLO_{final}$ tolerance. However, it did not improve growth under $LL_{12}$ or $HL_{1200}$ conditions, nor could it mediate $FL+_{final}$ tolerance (see Fig. 3 and Supplementary Fig. 2). This aligns with previous studies that reported a lethal phenotype in Pam68 deletion mutants exposed to alternating darkness and HL conditions[55]. Overexpression of WT Pam68 also enhanced $FLO_{final}$ tolerance, albeit to a lesser extent than in $pam68_{S113G}$ (see Supplementary Fig. 4). Total $Pam68_{S113G}$ protein levels decreased under $FLO_{final}$ conditions compared to LT Pam68 levels (see Fig. 5). Additionally, $pam68_{S113G}$ slightly stimulated the accumulation of PSII core and peripheral phycobililisome antenna proteins under $LL_{50}$ (see Fig. 5). However, as no growth improvement was observed under $LL_{12}$ conditions (see Supplementary Fig. 2), carbon assimilation, rather than light energy harvesting and conversion, may be the limiting factor for growth under LL conditions, as previously proposed[16,56].

Pam68 promotes the accumulation of PSII assembly intermediates RCa and RCb[40], as well as CP47 biosynthesis and chlorophyll ligand insertion[55]. Surprisingly, the $pam68_{S113G}$ mutant showed reduced *de-novo* biosynthesis of membrane proteins and decreased accumulation of PSII assembly intermediates, while dimeric PSII was found increased (see Fig. 5c, Supplementary Fig. 5). This effect aligns with the increased abundance of dimeric PSII observed in Pam68-depleted strains[40]. However both the apparent PSII quantum yield and PSII activity were significantly increased in $pam68_{S113G}$ cells (see Supplementary Fig. 6a, c), indicating increased PSII stability in $pam68_{S113G}$. The slower P700 oxidation and faster re-reduction indicate enhanced CEF activity (Supplementary Fig. 6d), although these effects can be partly attributed to reduced PSI abundance in $pam68_{S113G}$ cells grown under $LL_{50}$ conditions. A PAM68 orthologue (PAM68-LIKE) acts as an NDH-1 complex assembly factor in Arabidopsis[57]. As there is no second $pam68$ homologue in *Synechocystis*, it could be speculated that the *Synechocystis* Pam68 protein fullfils both functions, with the $Pam68_{S113G}$ mutation increasing NDH-dependent CEF activity. However, inactivation of Pam68 had no effect on NDH-1 assembly in *Synechocystis*[57], which suggests that this is not the case.

In sum, our findings suggest that $pam68_{S113G}$ is likely a gain-of-function mutation. Meanwhile, the precise molecular mechanism causing the depletion of mutant $Pam68_{S113G}$ protein levels under FL

conditions remains to be elucidated in future studies. The described $pam68$ gain-of-function mutation appears to be organism-specific, as a serine-to-glycine substitution at the homologous position of AtPAM68 did not increase FL tolerance when expressed in *Synechocystis* (see Fig. 4). Nevertheless, expressing the WT AtPAM68 protein, but not the mutant $AtPAM68_{S174G}$ protein, in the $pam68$ knock-out background significantly increased *Synechocystis* growth under $HL_{700}$ conditions. This suggests that, together with the glycine residue at position 174, substitutions in other positions of the Arabidopsis PAM68 protein confer $HL_{700}$ tolerance, likely due to evolutionary adaptation to the increased light intensities associated with a terrestrial light style. Therefore, while the Ser113/174 position is crucial and conserved for Pam68/PAM68 activity, the requirements for a gain of function under stressful light conditions have changed over evolutionary time.

RpaB is a redox-responsive OmpR-type transcription factor. It was first identified in *Synechocystis* due to its ability to alter the energy distribution from phycobilisomes to PSI compared to PSII[41]. RpaB regulates at least 137 promoters of protein-coding genes or operons, as well as 22 non-coding RNAs, including many genes involved in photosynthesis[50]. The $rpaB$ gene is essential[41,58], and orthologues have been identified in the plastid genomes of algae that also possess genes encoding phycobiliproteins[47]. RpaB binds to the high-light regulatory 1 (HLR1) sequence in PSI gene promoters[49], and under LL, it can function as either an activator or a repressor, depending on the position of the HLR1 sequence. Under HL, however, it loses binding activity. Consequently, RpaB can repress certain HL-inducible genes under LL and activate other genes, such as those encoding PSI subunits[48,59,60]. In *Synechococcus*, this involves the reversible phosphorylation of RpaB[58,61]. RpaB is also redox-regulated via a thiol switch; active dimers dissociate into less active monomers upon reduction by thioredoxin[62,63]. This links its function to the redox state of the photosynthetic electron transport chain. We identified two non-synonymous mutations in the $rpaB$ gene (Table 2). Notably, neither mutation affected the thiol switch residue (Cys59) or the likely phosphorylation site (Ser198)[64]. The $rpaB_{T183P}$ strains remained uniquely viable under $FL+_{final}$ and $HL_{1200}$, grew comparably to LT under $FLO_{final}$, and had a lower growth rate at $LL_{12}$ compared to LT (see Fig. 3, Supplementary Fig. 2). This suggests a trade-off: enhanced tolerance to HL in both continuous and fluctuating application at the expense of LL performance.

$RpaB_{T183P}$ levels were significantly increased under both $LL_{50}$ and $HL_{700}$. Previous studies have shown that $rpaB$ knock-down impairs growth under LL but promotes growth under HL conditions[49,65]. However, in our study, neither knock-down nor overexpression of $rpaB$ had positive effects on growth under $HL_{1200}$ and $FL+_{final}$ (see Supplementary Fig. 7), demonstrating that $rpaB_{T183P}$ confers additional functionality to $RpaB_{T183P}$, possibly by downregulation of PSII core and peripheral antenna proteins under elevated light intensities (see Fig. 6 and Supplementary Fig. 8). This aligns with observations from suppressor screens in *Arabidopsis thaliana pgr5* mutants, where increased FL sensitivity was overcome through mutational disruption of the photosynthetic electron transport chain to prevent PSI damage[66].

The $rpaB_{T183P}$ mutant strains exhibited a notable decrease in PC:Chl a ratios, particularly under $HL_{700}$ (see Supplementary Fig. 8), suggesting that increased FL tolerance is linked to reduced light energy harvesting capacity through peripheral antennas. This was accompanied by a significant reduction in phycobilisome and PSII fluorescence emission, consistent with previous findings in $rpaB$ knockdown strains[41]. Under $HL_{700}$, the mutant cells showed a marked decrease in qN[67] (see Supplementary Fig. 8), likely due to reduced PSII and phycobilisome levels. In contrast, under $LL_{50}$ conditions, the mutant displayed a reduced abundance of peripheral antenna proteins, decreased S2-to-S1 state transition, as well as an increased qN (see Fig. 6 and Supplementary Fig. 8). The latter is consistent with reports of increased OCP-dependent quenching of PSII in state 2

cells[68]. Fostered state 2 persistence could also explain the increased CEF activity around PSI[69] (see Supplementary Fig. 9).

In $rpaB_{T183P}$, PsaA accumulation was found to be slightly, yet non-significantly, increased under $LL_{50}$ and $HL_{700}$ conditions (see Fig. 6). At the same time, mRNA-seq data indicate the repression of most phycobilisome-related and PSI genes in $rpaB_{T183P}$ under both $LL_{50}$ and $HL_{700}$ conditions (see Fig. 7). This suggests that $RpaB_{T183P}$ may have lost some of its PSI-gene activating activity under LL and some of its PSI-gene repressing activity under HL. At the same time, PsaA protein levels no longer directly reflect mRNA levels. Alternatively, the observed changes to the transcriptome may be independent of altered binding of the mutated RpaB to the corresponding genes and may instead represent compensatory effects due to the physiological changes triggered by the $RpaB_{T183P}$ mutation. Moreover, reduced PSII levels under $HL_{700}$ appear to be functionally uncoupled from both transcription regulation and PSI levels. This is evident from the slight increase in PSII gene transcripts and the fact that PSII levels do not closely track changes in PSI levels. In contrast, PSI levels tend to follow alterations in PSII levels more closely[70,71]. Combined with the improved viability of the mutant under HL and FL conditions, this suggests that $rpaB_{T183P}$ is a gain-of-function mutation. This shifts the regulatory activity of RpaB from PSI to PSII and its peripheral antennas (see Supplementary Fig. 9). At the same time, it suppresses phycobilisome accumulation and S2-to-S1 transition (see Fig. 6 and Supplementary Fig. 8). This, together with the apparently increased CEF and its impact on photosynthetic control, protects PSI.

Given that RpaB-like proteins are conserved across cyanobacterial and some plastid phylogenies[47], their homologues in eukaryotic algae could potentially be targeted to enhance FL tolerance. However, because land plants lack phycobilisomes, RpaB-based tolerance is not feasible. Nevertheless, the rationale behind RpaB photoprotection could be mimicked by increasing the PSI/PSII ratio and downregulating antenna size.

Taken together, single amino acid exchanges in various cyanobacterial proteins can increase tolerance to FL. Two of these mutations, $pam68_{S113G}$ and $rpaB_{T183P}$, have been examined in greater detail in this study. This provides initial physiological insights into how these mutations modulate FL tolerance. Cross-species experiments with the Pam68 protein indicate that the FL tolerance function of the mutation found in *Synechocystis* is not conserved in land plant. Therefore, given that green algae are much closer relatives of flowering plants, it seems more practical to use ALE with green algae to identify adaptive mutations that might also function in flowering plants.

# Methods

### *Synechocystis* strains: generation and culture conditions

*Synechocystis* sp. PCC 6803 glucose-tolerant cells, referred to as "laboratory type" (LT) were kindly provided by Himadri Pakrasi (Washington University, St. Louis, USA). Previously described knock-out and knock-down mutants of *pam68* and *rpaB* were utilized[40,49]. Reconstruction of $sllO518_{A133V}$, $pam68_{S113G}$, $rpaB_{T183P}$ in the LT background employed established methodology[35] with plasmid vectors constructed using a pUC57-mini vector backbone derived from IMBB2.4-pUC57-mini kindly provided by Professor Neil Hunter (University of Sheffield). The fragment of codon-optimized *pam68* from *Arabidopsis thaliana* (AT4G19100.1) without cTP (*Atpam68*) was synthesized by Invitrogen GeneArt Strings (Thermo Fisher Scientific, MA, USA). Overexpression mutants of *pam68, rpaB,* and *Atpam68* were generated through homologous recombination using non-replicative vectors derived from IMBB2.4-pUC57-mini and pICH69822 (obtained from E. Weber, Icon Genetics GmbH, Halle, Germany), respectively. These constructs were assembled via Gibson assembly and targeted to the genomic neutral site *slr0168*, with *pam68* and *rpaB* coding sequences expressed under the control of the strong *psbA2* and *rbcL* promoters, respectively.

Cultures were typically grown under continuous illumination at 30 μmol photons m$^{-2}$ s$^{-1}$ of white fluorescent light (OSRAM HE28W/830 Lumilux warm white Hg fluorescent lamps) at 23 °C. This temperature aligns well with the average maximum temperature recorded in Oakland, California, over the year (https://weatherspark.com/y/541/Average-Weather-in-Oakland-California-United-States-Year-Round), where the original strain was isolated[72]. Liquid cultures were inoculated at an initial OD = 0.05 in BG11 photoautotrophic medium, with 5 mM glucose added for pre-transformation cultures. Growth was conducted in Multi-Cultivator MC 1000-OD devices, equipped with an AC-700 cooling unit and a warm-white LED panel (Photon System Instruments, Drasov, Czech Republic). For solid media growth, BG11 was supplemented with 0.75% (w/v) bacteriological agar.

### Adaptive evolution of *Synechocystis* under fluctuating light

Two FL adaptive evolution experiments were conducted using *Synechocystis*, relying on its natural mutation rates[35] to identify previously unknown adaptive alleles. The experiments began with six separate batch cultures derived from a *Synechocystis* LT stock culture. Each culture was grown in a 100 mL glass tube within a temperature-controlled water bath of a multicultivator. The propagation cycles involved 70 mL of medium with an initial $OD_{730nm}$ of 0.05, incubated under constant aeration at 23 °C with fluctuating warm-white LED illumination for 7–14 days. The light fluctuations were progressively intensified in both amplitude and frequency throughout the selection process.

The FL0 regime alternated between 1 min of high light (HL) and 5 min of low light (LL) throughout the entire adaptive laboratory evolution (ALE) protocol. The selection process began with five cycles of 50 μmol photons m$^{-2}$ s$^{-1}$ ($LL_{50}$) and 700 μmol photons m$^{-2}$ s$^{-1}$ ($HL_{700}$), followed by three cycles with varying LL and HL intensities. The final 12 cycles used a regime of 5 min at $LL_{12}$ and 1 min at $HL_{1200}$.

The FL+ regime also started with 1 min of HL followed by LL, but the LL period was progressively shortened from 5 min to 1 min. The initial cycle matched that of the FL0 regime, with subsequent cycles gradually reducing LL intensity and increasing HL intensity. The final 12 cycles alternated between 1 min of $LL_{12}$ and 1 min of $HL_{1200}$.

Both FL0 and FL+ experiments consisted of 20 selective cycles in total, resulting in three evolved batch cultures for each condition, labelled $FL0\_a_{20}$, $FL0\_b_{20}$, $FL0\_c_{20}$ and $FL+\_a_{20}$, $FL+\_b_{20}$, $FL+\_c_{20}$, respectively (Fig. 1a). Further details of the selective cycle protocols can be found in Table 1.

### Isolation of FL-adapted *Synechocystis* clones for genome re-sequencing

Single clones were isolated by plating dilutions ($10^{-6}$–$10^{-7}$) of $FL0\_a_{20}$, $FL0\_b_{20}$, $FL0\_c_{20}$ and $FL+\_\alpha_{20}$, $FL+\_\beta_{20}$, $FL+\_\gamma_{20}$, onto solid BG11 media. Isolation plates were incubated at 30 μmol photons m$^{-2}$ s$^{-1}$ continuous illumination and 23 °C for seven days. Representatives of mutant subpopulations were sampled by selecting different clones based on colony colour and size and isolated clones were grown on solid BG11 media for seven days.

Subsequently, clones for genome re-sequencing were selected as previously described[35] based on room-temperature fluorescence parameters measured by FluorCam 800MF (Photon Systems Instruments, Drasov, Czech Republic). For FL0 and FL+, $n = 66$ and $n = 72$ isolated clones were assessed. To capture the genetic variability within each batch culture, clones best representing the quartiles of the observed the $Fv^-/Fm^-$ distributions (i.e., two extreme and two intermediate values of PSII quenched quantum efficiency) were selected for whole-genome resequencing, totalling $n = 12$ for FL0 and FL+, respectively.

### *Synechocystis* genomic DNA extraction and sequencing

Genomic DNA for whole-genome sequencing was isolated from cell pellets of 10–30 mg fresh weight following the manufacturer's

protocol (EasyPure® Plant Genomic DNA Kit, TransGen Biotech Co., Ltd., Beijing, China). Cells were broken in EasyPure®lysis buffer using a 1:1 mixture of small glass beads (425–600 μm + 212–300 μm, Sigma Aldrich, St. Louis, MO, United States) and a TissueLyser II (QIAGEN, Hilden, Germany). DNA isolates were then subjected to agarose gel electrophoresis to assess structural integrity. The genomes of 24 monoclonal mutants (four per adapted batch culture) were then re-sequenced on the Illumina HiSeq platform (2× 150-bp paired-end reads) by NovoGene Ltd. (Cambridge, United Kingdom).

## Sequence data quality control and filtering

The previously published *Synechocystis* LT$_{t=0}$ genome assembly[35] served as the control for excluding background mutations; all mutations identified in FL-ALE were tracked relative to the LT$_{t=0}$ assembly.

Adapting previously described methodology[35], the quality of the WGS raw data was assessed using FastQC v0.11.9[73]. Pre-processing began with Cutadapt v4.1[74], which filtered out low-quality reads and removed sequences containing adaptor contamination or more than 10% undetermined bases ('N' bases). Rcorrector[75] was then used to perform *k*-mer correction on the filtered datasets, applying the default *k*-mer length setting. The resulting dataset, consisting of filtered and corrected reads, was used for subsequent mutation detection. Genome resequencing of the 24 single clones yielded an average coverage of $339 \pm 44/638 \pm 137/300 \pm 65/116 \pm 42/577 \pm 193$-fold for chromosome/pSYSM/pSYSA/pSYSG/pSYX, respectively (see Source Data).

## Variant analysis

The Breseq pipeline[76] were applied for the identification of potential mutations. The clean reads were aligned to the *Synechocystis* sp. PCC 6803 reference genome (ASM972v1) obtained from the NCBI database using bowtie2 v2.5.1[77]. The generated SAM alignment files were then used for variant calling. Breseq analysis was conducted in two distinct modes. The 'consensus' mode defined a mutation as 'fixed' when its frequency was ≥0.80, while considering a site as 'polymorphic' when the variant frequency ranged between 0.20 and 0.80. On the other hand, in the 'polymorphism' mode, a mutation was designated as 'fixed' at frequencies ≥0.95 and as 'polymorphic' if its occurrence spanned frequencies between 0.05 and 0.95. Identified variants are listed in Source Data. For an overview of fully segregated, protein-affecting mutations identified in FL-ALE strains, see Table 2.

## Phylogenetic analysis

The phylogenetic analysis was carried out considering 101 polymorphic sites representing all deviations from the reference genome (ASM972v1) with a 100% frequency (i.e., identified as fully segregated in at least one sample). IQ-TREE multicore version 2.2.6[78] was used to perform the subsequent phylogenetic analysis. Briefly, the model selection method was applied with default parameters to identify the most suitable model for the data set. Afterwards, the selected model (according to Bayesian Information Criterion "BIC" values), that is Kimura 2 Parameter (K2P) with equal frequencies, was applied to infer the maximum likelihood phylogenetic relationships between the samples. The bootstrapping method was applied to validate the generated tree with 500 replicates. The resulting phylogenetic tree was generated using CLC Main Workbench (QIAGEN, Venlo, Netherlands).

## Pigment extraction and quantification, determination of phycocyanin:chlorophyll ratios

Chlorophyll *a* (Chl *a*) and total carotenoids (Cars), were extracted and quantified as previously described[35]. Molar ratios of the peripheral antenna pigment phycocyanin (PC) to core antenna pigment Chl a in cultures seven days past inoculation were estimated as previously described[35].

## Protein extraction, detection and quantification

Cells were collected from a 3-mL suspension at OD$_{730nm}$ = 10 by gentle centrifugation, and the pellets were snap-frozen in liquid N$_2$ and stored at −80 °C. The cell pellets were then homogenized and lysed in 600 μL of homogenization buffer (0.4 M sucrose, 10 mM NaCl, 5 mM MgCl$_2$, 20 mM Tricine, adjusted to pH 7.9 with HCl), supplemented with protease inhibitors (cOmplete™ Mini EDTA-free Protease Inhibitor Cocktail, Roche AG, Basel, Switzerland) and approximately 300 μL of a glass bead mixture. Lysis was performed using a mixer mill (MM 400, RETSCH, Haan, Germany) with five cold-lysis cycles (5 min at 30 Hz). After centrifugation at 4 °C, the supernatant was collected, and protein concentration was estimated using Bradford (ROTI-Quant, Carl Roth, Karlsruhe, Germany) and BCA (Pierce™ BCA Protein Assay Kit, Thermo Fisher Scientific, Waltham, MA, USA) protein assays. Chl a concentration was estimated as described above. Samples were stored at −20 °C until further processing.

For SDS-PAGE, protein extracts (equal Chl a content) were mixed with 5× SDS loading dye, denatured (37 °C, 60 min) and size-separated on 10% Tris-Tricine gels. Phycocyanin and allophycocyanin were quantified by recording fluorescence ($\lambda_{emission} \geq 600$ nm) directly from the gels[79] (Fusion FX imaging system, Vilber, Collégien, France; excitation wavelength of $\lambda_{excitation} = 530$ nm). Image analysis and signal quantification were conducted using ImageJ[80]. Proteins were then transferred to PVDF membranes (Immobilon-PSQ, Millipore, Burlington, MA, USA) via electroblotting. Membranes were stained with Coomassie Brilliant Blue (CBB) for loading control, then de-stained before immunodetection.

For specific protein detection, membranes were cut or used whole, blocked with 1.5% (w/v) BSA in TBST, and incubated with primary antibodies against PsaA, PsbA, and AtpB (Agrisera, Vännäs, Sweden), Pam68 (kindly provided by Prof. Dr. Jörg Nickelsen, LMU Munich, Germany), and RpaB (PhytoAB, San Jose, CA, USA). After overnight incubation with primary antibodies at 4 °C and 2-hour incubation at room temperature with horseradish-peroxidase coupled secondary antibodies, chemiluminescence was detected using Super-Signal™ West Pico PLUS chemiluminescent substrate (Thermo Fisher Scientific, Waltham, MA, USA) and imaging system (Fusion FX imaging system, Vilber, Collégien, France). Signal quantification was performed using ImageJ software[80].

## Preparation of the membrane fraction and analysis of proteins by clear native PAGE

To isolate the cellular membranes, the cells were disrupted using zirconia–silica beads in a Precellys Evolution tissue homogeniser (Bertin Instruments, France). The membrane and soluble fractions were then separated by centrifugation at $36,000 \times g$ for 20 min. The membranes were then resuspended in buffer A (25 mM MES/NaOH, pH6.5, 10 mMCaCl$_2$, 10 mM MgCl$_2$, 25% glycerol) and, after measuring the Chl concentration, solubilised with β-dodecyl-maltoside (DDM, final concentration 1% (w/v)) and analysed using two-dimensional PAGE consisting of clear native (CN) PAGE in a 4–14% gradient gel[81] and SDS-PAGE in a denaturing 16–20% gradient gel containing 7 M urea (2D-CN/SDS-PAGE). For autoradiography, the gels were stained with Coomassie Blue (CBB), destained, dried, and exposed to a phosphor-imager plate for 144 h.

For protein detection, the gels were stained with SYPRO Orange and subsequently transferred to a polyvinylidene fluoride (PVDF) membrane. The primary antibodies used in the study were raised in rabbits against the following: (i) D1 (residues 58–86 of the spinach D1 polypeptide); (ii) CP47 (residues 380–394 of the barley polypeptide); (iii) CP43 (Agrisera catalogue no. AS11 1787); and (iv) PAM68 (residues 1–63 of the *Synechocystis* polypeptide[40]). The antibodies were used sequentially in the following order: PAM, CP43, CP47 and D1. The blots were developed using an anti-rabbit secondary antibody conjugated

with horseradish peroxidase (Merck, USA), alongside a chemiluminescence substrate (Immobilon Crescendo, Merck, USA).

## Radioactive labelling

Radioactive pulse labelling of the cells was performed at an intensity of 500 μmol photons $m^{-2} s^{-1}$ and a temperature of 30 °C, using a mixture of [$^{35}$S]Met and [$^{35}$S]Cys (Hartmann Analytic Gmbh, Braunschweig, Germany) as previously described[82].

## Low-temperature fluorescence spectrometry

Low-temperature fluorescence spectra of Chl a and phycobiliproteins were recorded using a HORIBA Fluoromax Plus FL-1013 spectrofluorometer (HORIBA Jobin Yvon GmbH, Oberursel, Germany). Cultures grown in MC photobioreactors under the specified light conditions were transferred into glass capillaries (Hilgenberg GmbH, Malsfeld, Germany) directly from the cultivation device at 7 days past inoculation and immediately snap-frozen in liquid nitrogen. Samples were stored in a light-occluding casing at −80 °C for 1–2 weeks and measured in a single batch. Measurements were conducted at 77 K in a dewar filled with liquid nitrogen, using a signal integration time of 0.2 s $nm^{-1}$ and a detection bandwidth of 1 nm. Fluorescence signals were detected through a 500 nm long-pass filter upon excitation at 435 nm ($\lambda exc$ = 435 nm) to stimulate Chl a absorption and at 600 nm ($\lambda_{exc}$ = 600 nm) to selectively excite phycobiliproteins.

Key fluorescence peaks were analysed as ratios derived from relevant peaks. Relevant peaks corresponded to phycocyanin (PC; F645), allophycocyanin (APC; F662), Chl a in PSII core antenna CP43 (CP43; F685), Chl a in PSII core antenna CP47 (CP47; F695), and Chl a in PSI (PSI; F725) as described[35].

For the state transition analysis, the cells were cultivated for seven days at 23 °C in a multi-cultivator. The cultures were transferred directly from multi-cultivator into NMR capillaries and exposed to blue light (150 μmol photons $m^{-2} s^{-1}$) for State 1 and to darkness for State 2 for 15 min. Then the capillaries were then frozen in liquid nitrogen and stored at −80 °C until testing. The measurements were performed as described above.

## Determination of PSII quantum yield parameters

Apparent PSII quantum yield was measured as Fv⁻/Fm⁻[38] at room temperature as previously described[35] using a FluorCam 800MF (Photon Systems Instruments, Drásov, Czech Republic). Fv⁻/Fm⁻ was measured in cells grown for 7 days at 23 °C and $LL_{20}$ on BG11 solid media after single-clone isolation, or cells collected from cultures grown at $LL_{50}$ in multi-cultivators and subsequently normalized to $OD_{730nm}$ = 10 of which 10 μL droplets were placed on BG11-agar. After overnight acclimation at $LL_{20}$ and 23 °C, plates were dark-incubated for one hour before measurements. Instrument sensitivity was manually adjusted (10–20%) before measuring. For samples under HL, suspensions were collected from the liquid cultures after 7 day cultivation and prepared as described previously[8].

## P700 redox kinetics measurements

The cells for the P700 measurements were cultivated for seven days at appropriate light conditions and 23 °C in a Multi-Cultivator. Cells were harvested by a 4-min centrifugation at 2800 $g$ and 25 °C, washed twice in BG11, and adjusted to $OD_{730 nm}$ = 5. Cell suspensions were aliquoted into 2-ml fractions and incubated in the dark overnight (≥16 h) at 25 °C and 120 rpm. Oxidation and re-reduction of P700 was analysed by measuring P700 absorbance at 820 nm relative to 870 nm using the DUAL-PAM 100 instrument (Walz, Effeltrich, Germany), as previously described[35,83]. The measurement routine involved 3 s in the dark, 60 s of FR light, and 30 s in the dark at measuring light intensity 4, FR light intensity 3 (38 μmol photons $m^{-2} s^{-1}$), acquisition rate 200 $s^{-1}$, and high gain (5) and damping (1 ms). Curves were normalised by equating the absorbance baseline

(average absorbance over 3 s of dark before onset of FR) to 0, and equating the absorbance maximum during 60 s FR illumination to 1. Rate constants (time required to reach 50% P700 oxidation or re-reduction) were acquired from normalised curves by extracting the time points at which Δabs exceeded 0.5.

## Oxygen evolution and respiration measurements

Oxygen evolution was measured using an Oxytherm+ P Clark-type oxygen electrode (Hansatech Instruments, King's Lynn, UK) in non-disturbed *Synechocystis* cultures cultivated for seven days at 23 °C in a multi-cultivator. PSII activity was estimated as the steady-state $O_2$ evolution rate under saturating light at 23 °C in the presence of 0.5 mM DCBQ (1,4-benzoquinone) and 1 mM $K_3Fe(CN)_6$. Respiration was estimated in terms of the steady-state $O_2$ consumption rate over 5 min in the dark. The oxygen evolution rate and respiration rates were normalized to the $OD_{730nm}$ or Chl a content, respectively, for each sample.

## Determination of coefficient of non-photochemical quenching at room temperature

To determine the coefficient of non-photochemical quenching at room temperature, fast fluorescence kinetics were measured using a FluorCam 800MF (Photon Systems Instruments, Drásov, Czech Republic). Cells collected from cultures grown at 50 μmol photons $m^{-2} s^{-1}$ in MC were normalized to $OD_{730nm}$ = 10, and 10 μL droplets were evenly distributed on BG11-agar. After overnight acclimation at 30 μmol photons $m^{-2} s^{-1}$ and 23 °C, plates were dark-incubated for one hour before measurements. Instrument sensitivity was manually adjusted (20–30%) before measuring fluorescence kinetics under incremental red-orange illumination (Fo 5 s > Fm pulse 0.8 s > AL 10 % 60 s > Fm pulse 0.8 s > AL 20 % 60 s > Fm pulse 0.8 s > AL 40 % 60 s > Fm pulse 0.8 s > AL 60 % 60 s > Fm pulse 0.8 s > AL 80 % 60 s > Fm pulse 0.8 s > AL 100 % 60 s > Fm pulse 0.8 s > END) and induction-relaxation regime (Fo 5 s > Fm pulse 0.8 s > dark 10 s > AL 100 % 60 s > dark relaxation 60 s; 9 Fm pulses during AL and dark relaxation, first Fm pulse 9 s after AL onset).

Actinic light at 100% intensity approximated 215 μmol photons $m^{-2} s^{-1}$ ($\lambda_{max}$ = 625 nm), while saturating pulses (Fm pulses; ~1200 μmol photons $m^{-2} s^{-1}$) were provided by cold-white 6500 K LEDs. Quenching parameters were computed using FluorCam7 (Photon Systems Instruments).

## RNA extraction and transcriptome analysis

For RNA extraction, 30 ml of cell culture ($OD_{730nm}$ ~ 1.0) was harvested from LT and $rpaB_{T183P}$ cells, which were grown under constant $LL_{50}$ and $HL_{700}$ conditions in multi-cultivators at 23 °C. The cells were then centrifuged and resuspended in TRIzol (Invitrogen, Carlsbad, California, USA). After incubation of cells in the TRIzol at 65 °C for 15 min, the RNA was extracted using phenol-chloroform and finally precipitated with isopropanol overnight at −20 °C. The RNA was treated with the TURBO DNA-free™ kit (Invitrogen, MA, USA) to remove genomic DNA, and then purified using Direct-zol™ RNA MiniPrep Plus columns (Zymo Research, Irvine, California, USA). Ribosomal RNA depletion and RNA-seq library generation were performed by Novogene Biotech (Beijing, China) using standard Illumina protocols. The RNA-seq libraries were sequenced on an Illumina HiSeq 2500 system (Illumina, San Diego, Calif. USA) using a 150 bp paired-end sequencing strategy.

Data quality was checked using FastQC (version 0.12.1) and cleaned using fastp (version 1.0.1). Paired-end reads were mapped to the *Synechocystis* sp. PCC6803 genome (GCF_000009725.1) using HISAT2 (version 2.2.1). Read counts were quantified using featureCounts (version 2.1.1). The rest of the analysis was performed in R (version 4.5.2.) via Rstudio (version 2025.09.2, build 418). DESeq2 (1.48.2) was then used to identify differentially expressed genes.

## Statistics and boxplot description

Statistical analyses were conducted using two-tailed Student's $t$ tests in Microsoft Excel. Post-hoc Bonferroni–Holm corrections for multiple comparisons were performed using astatsa when significant differences among groups were detected via one-way ANOVA at https://astatsa.com/.

Box plots depicting data point distributions were created in Microsoft Excel. The horizontal middle lines represent inclusive medians, crosses indicate mean values, boxes denote the second and third quartiles, whiskers extend to the first and fourth quartiles, and points beyond the whisker ranges indicate outliers exceeding 1.5 times the interquartile range.

## Structural predictions

AlphaFold3 (https://alphafoldserver.com/) was used to predict the tertiary structures of Sll0518, PAM68 and RpaB, as well as their variants with point mutations. The top-ranked predictions from AlphaFold3 were selected for further calculations and comparisons using UCSF Chimera X (University of California, San Francisco).

## Reporting summary

Further information on research design is available in the Nature Portfolio Reporting Summary linked to this article.

## Data availability

Vector sequences are listed in Source Data. DNA-Seq and RNA-Seq data are available from the NCBI SRA database under Project number: PRJNA1228058 and Project number: PRJNA1372014. Source data are provided with this paper.

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

## Acknowledgements

We thank Prof. Jörg Nickelsen (LMU Munich, Germany) for providing the Pam68 antibody and Prof. Yukako Hihara (Saitama University, Japan) for sharing the RpaB knock-down mutant (*rpaBkd*). This study is supported by the Deutsche Forschungsgemeinschaft (grant TRR175 [project number: 270050988] to D.L.), the European Research Council (ERC Synergy Grant "PhotoRedesign" [project number: 854126], to D.L. and J.K.), the Czech Ministry of Education (project PHOTOMACHINES, CZ.02.01.01/ 00/22_008/0004624 to D.S. and J.K.) and the Deutscher Akademischer Austauschdienst (DAAD, to T.F.-G.).

## Author contributions
D.L., M.D. and T.F.-G. conceived the project. D.L. provided the funding. D.L., T.F.-G., W.C. and J.K. designed all experiments, with support from M.D. T.F.-G., W.C., M.Z. and D.S. performed the experiments. W.C. and E.M.A.-S. performed the DNA and RNA sequencing analyses and interpretation of the results. D.L. wrote the manuscript with the support of M.D., W.C. and T.F.-G. All authors read and approved the final manuscript.

## Funding

## Competing interests
The authors declare no competing interests.
