## [Transparent Peer Review file · Nature Communications]

Improving tolerance to fluctuating light through adaptive laboratory evolution in the cyanobacterium *Synechocystis*

Corresponding Author: Professor Dario Leister

Version 0:

Reviewer comments:

Reviewer #1

(Remarks to the Author)

If anything this paper has already been over-reviewed already, so I will make no suggestions for further work, only assessing the relative value of replies to comments from Reviewer 3, whom I have agreed to replace, on my specialist subject of photosynthetic measurements.

Comment 3.1 and reply. The reviewer questions the interpretation of increased CEF in pam68S113G, and suggests adding measurements of P700 re-reduction to those of oxidation rate. The authors comply well with this request and tone down their interpretation. The data are difficult to explain, and not key to the major findings, so the authors are to be commended for keeping them, aiding transparency and dissemination of data to the community. This data is mainly discussed in lines 244-254.

I do have some comments here. The change relative to LT is quite dramatic. The authors don't state in the legend for how long the cultures were incubated in the dark, but even with very extensive dark incubation some degree of respiration must persist or the cell would die, and this is highly likely to be consistent between LT and pam68S113G. I would therefore interpret the P700 re-reduction and oxidation data more as the contribution of the respiratory chain (and interchain electron carriers) to P700 kinetics. This is interesting in and of itself. I see that the authors measured chlorophyll content of their strains, but cannot find a direct comparison, apart from the mention that "FL+ strains exhibited higher growth rates and increased cell density compared to FLO cultures, although with lower chlorophyll content" on line 81. If the pam68S113G has lower chlorophyll (most of which is at PSI in cyanobacteria), then it would be logical that the relatively higher respiratory electron supply through the cytochrome bf would slow oxidation and speed up re-reduction of P700 relative to the LT. Minor correction suggested

I actually disagree with the statement made by the authors in line 253 "The increased CEF could contribute to enhanced FL tolerance by alleviating PSI acceptor site limitation". You can find similar statements in many other papers from many other groups, so it is not heresy. However, I believe it is wrong because CEF is by nature a zero sum game, not a valve. Electrons will return to P700, compounding any acceptor limitation. Minor correction suggested

Comment 3.2 and reply. The reviewer complains about phrasing and points out PSII activity relative to the wt depends on normalisation (lower vs cell count, higher vs chl a). I agree that this suggest lower Chl a in the mutant (see my comment above). The reviewer says that Fv/Fm values are low for the LT control, but I disagree with the reviewer. 0.39 is respectable for a cyanobacteria. They request actual values for O₂ evolution and Fv/Fm, which the authors supply. Good.

Comment 3.3 and reply. This again relates to measuring CEF - a complex and controversial topic in cyanobacterial photosynthesis research! They suggest toning down the conclusions and adding re-reduction rate, which the Authors now supply. The authors are to be commended for adding the extra data and discussing it. I still think that the data might be explained by lower active P700 relative to donor supply (see above).

Comment 3.4 and reply. The reviewer asks why are the data expressed as PSI/PSII ratio not F695/F730 and suggest

involvement of the alternative antenna protein IsiA. The authors sensibly ignore the comment on IsiA – many things can contribute to fluorescence in this region, and explain the rationale for the labelling well. Good

Comment 3.5 and reply. The reviewer questions the validity of the ratios expressed for PC/APC and PSII/APC, and the source data. The authors supply the source data and explain the rationale well. Good

Comment 3.6 and reply. The reviewer questions what could contribute to quenching of chlorophyll fluorescence at low light. The authors give a sensible reply explaining state transitions, which are also outlined in the text, in line with the role of RpaB in state transitions. Good

Comment 3.7 and reply. the reviewer requests further methodological detail, which the authors supply. In addition the reviewer asks why FluorCam was used for Fv/Fm, but dual PAM for P700. I share the reviewers concern as FluorCam measurements tend to be less accurate. The authors explain it was to keep things consistent between isolation protocols and final measurements. This does not sit well with me if they have the accurate Dual-PAM data available. They say they observed the same trends when remeasuring with the dual PAM, but would prefer to keep things consistent through the paper. I understand that it might not look as nice if the absolute values are changing between the isolation and characterisation figures, but would like to see the data from the more accurate method included somewhere. Minor correction suggested

If anything this paper has already been over-reviewed already, so I will make no suggestions for further work, only assessing the relative value of replies to comments from Reviewer 3, whom I have agreed to replace, on my specialist subject of photosynthetic measurements.

1. Comment 3.1 and reply. The reviewer questions the interpretation of increased CEF in *pam68S113G*, and suggests adding measurements of P700 re-reduction to those of oxidation rate. The authors comply well with this request and tone down their interpretation. The data are difficult to explain, and not key to the major findings, so the authors are to be commended for keeping them, aiding transparency and dissemination of data to the community. This data is mainly discussed in lines 244-254.

Reply 1. Many thanks

2. I do have some comments here. The change relative to LT is quite dramatic. The authors don't state in the legend for how long the cultures were incubated in the dark, but even with very extensive dark incubation some degree of respiration must persist or the cell would die, and this is highly likely to be consistent between LT and *pam68S113G*. I would therefore interpret the P700 re-reduction and oxidation data more as the contribution of the respiratory chain (and interchain electron carriers) to P700 kinetics. This is interesting in and of itself. I see that the authors measured chlorophyll content of their strains, but cannot find a direct comparison, apart from the mention that "FL+ strains exhibited higher growth rates and increased cell density compared to FLO cultures, although with lower chlorophyll content" on line 81. If the *pam68S113G* has lower chlorophyll (most of which is at PSI in cyanobacteria), then it would be logical that the relatively higher respiratory electron supply through the cytochrome *b₆f* would slow oxidation and speed up re-reduction of P700 relative to the LT. Minor correction suggested

Reply 2. The incubation time in the dark is mentioned in the Methods section (overnight, i.e. ≥ 16 h). This information has now been also inserted in the legend of Fig. S6. We address the suggestion of the reviewer and write now: "Increased CEF may support improved FL tolerance by enhancing Δ pH formation and thereby restricting electron flow through cytochrome *b₆f* into PSI (photosynthetic control), while temporarily alleviating PSI acceptor site limitation upon HL onset during FL cycles. However, under LL₅₀ conditions, *pam68S113G* likely contains less PSI than LT, as suggested by its $\sim 20\%$ lower chlorophyll a content (Source Data S5). Consequently, a relative increase of respiratory electron input via cytochrome *b₆f* may also contribute to the slower P700 oxidation and faster re-reduction relative to LT in LL₅₀ samples. Consistent with this, chlorophyll content does not differ significantly between genotypes under FLO conditions, and differences in P700 kinetics are correspondingly less pronounced."

3. I actually disagree with the statement made by the authors in line 253 "The increased CEF could contribute to enhanced FL tolerance by alleviating PSI acceptor site limitation". You can find similar statements in many other papers from many other groups, so it is not heresy. However, I believe it is wrong because CEF is by nature a zero sum game, not a valve. Electrons will return to P700, compounding any acceptor limitation. Minor correction suggested.

Reply 3. Correct. See above. We write now: "Increased CEF may support improved FL tolerance by enhancing Δ pH formation and thereby restricting electron flow through cytochrome *b₆f* into PSI (photosynthetic control), while temporarily alleviating PSI acceptor site limitation upon HL onset during FL cycles. Later in the discussion we write: "This, together with the apparently increased CEF and its impact on photosynthetic control, protects PSI."

4. Comment 3.2 and reply. The reviewer complains about phrasing and points out PSII activity relative to the wt depends on normalisation (lower vs cell count, higher vs chl a). I agree that this suggest lower Chl a in the mutant (see my comment above). The reviewer says that Fv/Fm values are low for the LT control, but I disagree with the reviewer. 0.39 is respectable for a cyanobacteria. They request actual values for O₂ evolution and Fv/Fm, which the authors supply. Good.

Reply 4. Thank you.

5. Comment 3.3 and reply. This again relates to measuring CEF - a complex and controversial topic in cyanobacterial photosynthesis research! They suggest toning down the conclusions and adding re-reduction rate, which the Authors now supply. The authors are to be commended for adding the extra data and discussing it. I still think that the data might be explained by lower active P700 relative to donor supply (see above).

Reply 5. We write now in the Discussion section: "The slower P700 oxidation and faster re-reduction indicate enhanced CEF activity (**Supplementary Fig. 6d**), although these effects can be partly attributed to reduced PSI abundance in *pam68_{S113G}* cells grown under LL₅₀ conditions."

6. Comment 3.4 and reply. The reviewer asks why are the data expressed as PSI/PSII ratio not F695/F730 and suggest involvement of the alternative antenna protein IsiA. The authors sensibly ignore the comment on IsiA – many things can contribute to fluorescence in this region, and explain the rationale for the labelling well. Good

Reply 6. Thank you.

7. Comment 3.5 and reply. The reviewer questions the validity of the ratios expressed for PC/APC and PSII/APC, and the source data. The authors supply the source data and explain the rationale well. Good

Reply 7. Thank you

8. Comment 3.6 and reply. The reviewer questions what could contribute to quenching of chlorophyll fluorescence at low light. The authors give a sensible reply explaining state transitions, which are also outlined in the text, in line with the role of RpaB in state transitions. Good

Reply 8. Thank you

9. Comment 3.7 and reply. the reviewer requests further methodological detail, which the authors supply. In addition the reviewer asks why FluorCam was used for Fv/Fm, but dual PAM for P700. I share the reviewers concern as FluorCam measurements tend to be less accurate. The authors explain it was to keep things consistent between isolation protocols and final measurements. This does not sit well with me if they have the accurate Dual-PAM data available. They say they observed the same trends when remeasuring with the dual PAM, but would prefer to keep things consistent through the paper. I understand that it might not look as nice if the absolute values are changing between the isolation and characterisation figures, but would like to see the data from the more accurate method included somewhere. Minor correction suggested

Reply 9. We write now: *In-vivo* fluorimetry (FluorCam) also indicated an increase in Fv/Fm in *pam68_{S113G}* mutants adapted to LL₂₀ (+21%; p=4.55x10⁻¹⁶) and FLO_{final} (+118%; p=2.75x10⁻²¹) (**Supplementary Fig. 6c**), further suggesting that *pam68_{S113G}* enhances PSII performance. A similar trend was observed using a DualPam fluorometer (Source Data **S5**)